# MEDIATOR: MEMORY-EFFICIENT LLM MERGING WITH LESS PARAMETER CONFLICTS AND UNCERTAINTY BASED ROUTING

## ABSTRACT

Model merging aggregates Large Language Models (LLMs) finetuned on different tasks into a stronger one. However, parameter conflicts between models leads to performance degradation in averaging. While model routing addresses this issue by selecting individual models during inference, it imposes excessive storage and compute costs, and fails to leverage the common knowledge from different models. In this work, we observe that different layers exhibit varying levels of parameter conflicts. Building on this insight, we average layers with minimal parameter conflicts and use a novel task-level expert routing for layers with significant conflicts. To further reduce storage costs, inspired by task arithmetic sparsity, we decouple multiple fine-tuned experts into a dense expert and several sparse experts. Then, we select and merge appropriate experts based on the task uncertainty of the input data. We conduct extensive experiments on both LLaMA and Qwen with varying parameter scales, and evaluate on real-world reasoning tasks. Results demonstrate that our method consistently achieves significant performance improvements while requiring less system cost.

## 1 INTRODUCTION

Finetuning Large Language Models (LLMs) enables them to adapt to downstream applications including sentiment analysis (Sun et al., 2023), text summarization (Fang et al., 2024), mathematical reasoning (Ruis et al., 2024), code writing (Jiang et al., 2024a), roleplay chatting (Chen et al., 2025) so on. Open-source platforms such as Huggingface (Wolf et al., 2019) facilitate access to a diverse array of highly trained expert models with varying capabilities. Because the computational resources are scarce and implementing green computing (Stojkovic et al., 2024), the community is increasingly interested in how to merge these models to create a superior LLM that retains the strengths of finetuned ones without retraining (Yang et al., 2024a).

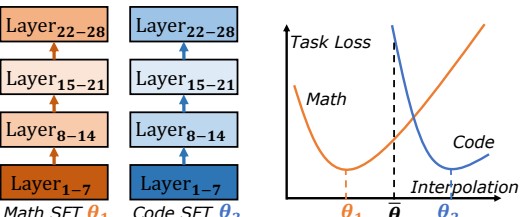

Figure 1: Knowledge conflict across finetuned LLMs and math and code dataset. Deeper color means larger parameter conflicts. And it is difficult for the linear averaged model to achieve low loss of both tasks.

Current merging strategies generally fall into two categories, each with distinct limitations. The first is model averaging (Yang et al., 2024a; Thennal et al., 2024), which computes weighted averages of parameters to synthesize collective knowledge. However, this approach suffers from parameter conflicts arising from diverse finetuning tasks, leading to performance degradation as shown in Figure 1. The second direction is model routing (Lu et al., 2024a; He et al., 2024a; Wei et al., 2024a), which maintains multiple models and selects the best one during inference. While this avoids parameter conflicts, it incurs significant computing and storage costs due to the necessity of maintaining all finetuned models in memory, and loses aggregating shared knowledge into one model. This dichotomy motivates us to rethink the following question:

*How to merge common and unique knowledge from various finetuned models while simultaneously avoiding parameter conflicts and minimizing system costs?*

To address this, we first investigate the nature of parameter conflicts. By employing sign consistency between different task arithmetics to measure conflict levels, we discover that conflicts are not uniform: front and last layers tend to exhibit the highest levels of conflict (noise), while central layers demonstrate comparatively lower levels (Section 3). This insight drives our proposed framework, Mediator, which employs an adaptive merging strategy. We average layers with lower conflict levels to capture common knowledge (He et al., 2024b), while layers with significant conflicts are treated as distinct experts to be routed during inference (Section 4.1). This hybrid approach effectively preserves unique task-specific knowledge without dilution.

However, the "adaptive merging" strategy introduces a new challenge: maintaining multiple expert layers for the high-conflict regions remains memory-intensive. To resolve this storage bottleneck, we introduce expert decomposition. Leveraging the high sparsity of task arithmetics (Yadav et al., 2023a), we decouple finetuned models into a shared dense base and lightweight, sparse task-specific components ($\Delta_\tau$) (Ilharco et al., 2023; Tang et al., 2024a) (Section 4.2). By storing only one dense model and $N$ sparse experts, we reduce storage requirements from 50% to 7%, making the system highly efficient.

With these sparse experts available, the challenge shifts to efficient selection. Existing token-level routing (Sukhbaatar et al., 2024a; Jiang et al., 2024b) often fragments the context, which is suboptimal given that LLMs are finetuned on complete sentences. Therefore, we propose a task-level expert router (Section 4.3). This mechanism selects the appropriate sparse experts based on the input prompt, better preserving the integrity of task-specific knowledge.

Furthermore, a static task-level router may struggle with complex or out-of-distribution (OOD) samples. To address this, we incorporate uncertainty-based routing. Mediator estimates task uncertainty for input data. For ambiguous or OOD inputs, it intelligently merges multiple sparse experts to create a composite expert on the fly (Section 4.3). This allows the unified model to adapt dynamically to the characteristics of the input data.

Finally, to validate that our system preserves high-level capabilities, we evaluate Mediator using Chain-of-Thought (CoT) enhanced datasets (Wei et al., 2022; Guo et al., 2025) (Section 5). Our evaluations show that Mediator achieves minimal performance degradation ($0.06\% \sim 0.3\%$) while enabling a model comparable to a 7B $\times$ 4 LLM ensemble to run effectively on a single consumer-grade RTX 4090 GPU (Appendix I.4).

Our main contributions can be summarized as follows:

- We quantify layer-wise parameter conflicts (Section 3) and propose Mediator, an adaptive merging framework that averages low-conflict layers to share common knowledge while routing high-conflict layers to avoid interference (Section 4.1).

- To solve the storage overhead of routing, we introduce expert decomposition (Section 4.2), separating models into one dense base and multiple sparse experts. This achieves a high compression ratio while maintaining accuracy.

- We develop an uncertainty-based routing mechanism (Section 4.3) that performs task-level selection for standard inputs and dynamic expert merging for OOD data. Experiments on LLaMA and Qwen with CoT datasets demonstrate that Mediator outperforms existing methods in both accuracy and system efficiency (Section 5).

## 2 PRELIMINARY AND RELATED WORKS

### 2.1 LANGUAGE MODELING AND LLM FINETUNING

**Task Data Distribution.** Given a set of different downstream tasks $\mathcal{T}$, based on the sampling task $\tau \in \mathcal{T}$, the pretraining data sample is a sequence $x_{1:T}$ of tokens with the maximum length $T$ generated from a distribution $p_\tau = p(x_{1:T}|\tau) = p(o_1, \ldots, o_T|\tau)$ (Xie et al., 2022; Wies et al., 2023; Hahn and Goyal, 2023; Li et al., 2024a). We define the pretraining data is sampled from

$p(x|\mathcal{T}^\star) = \int_{\tau^\star \in \mathcal{T}^\star} p(o_1, \ldots, o_T|\tau) p(\tau^\star) d\tau^\star$. Each token $o$ is sampled from a vocabulary $\mathbb{O}$. And both ($\mathcal{T}$ and $\mathcal{T}^\star$ belong to a large task family $\Omega$, i.e. $\mathcal{T}, \mathcal{T}^\star \subset \Omega$.

**Language Modeling.** Current LLMs (Brown et al., 2020; Touvron et al., 2023; Xie et al., 2022) usually utilize the next word prediction as the language modelling, which predicts the next token $x_t$ given the previous tokens $x_{1:t-1}$ for all $t = 1, \ldots, T$. Formally, a LLM parameterized by $\theta$ is a distribution $f_\theta(x_t|x_{1:t-1})$. And it is pretrained on a huge corpus sampled from the pretraining distribution $p(x|\mathcal{T}^\star)$ (Xie et al., 2022).

**Finetuning LLM.** Normally, for each downstream task $\tau \in \mathcal{T}$, finetuning LLM is to minimize the cross-entropy loss function as $L_{\text{CE}}(\theta, \tau) = -\sum_{t=1}^T \mathbb{E}[p_\tau(x_t|x_{1:t-1}) \cdot \log f_\theta(x_t|x_{1:t-1})]$. After finetuning, the model parameters $\theta$ are updated to $\theta_\tau$.

## 2.2 MODEL MERGING

Given finetuned task-specific LLMs $\{\theta_1, \theta_2, \ldots, \theta_{n_\tau}\}$ finetuned on task set $\mathcal{T}$, where $n_\tau = |\mathcal{T}|$, model merge aims to find a unified model parameterized by $\phi$ that can achieve the low loss on all tasks $\mathcal{T}$ as following

$$\min_\phi L_{\text{CE}}(\phi, \mathcal{T}) = \frac{1}{n_\tau} \sum_{\tau \in \mathcal{T}} L_{\text{CE}}(\phi, \tau). \tag{1}$$

Different from training $\phi$ that can be optimized towards any direction, model merging aims to exploit combining $\{\theta, \theta_1, \theta_2, \ldots, \theta_{n_\tau}\}$ to obtain the $\phi$. The current model merging methods include following two categories.

**Model Averaging.** Averaging parameters to fuse the knowledge from different finetuned models is straightforward. Mathematically, averaged model is $\phi = \sum_{\tau \in \mathcal{T}} w_\tau \theta_\tau$, in which $w_\tau$ is the averaging weight and $\sum_{\tau \in \mathcal{T}} w_\tau = 1$. Because different model parameters have different importance on downstream tasks, some works (Kirkpatrick et al., 2017; Sun et al., 2024; Dong et al.) assign larger weights to more important parameters. Current methods usually utilize Taylor expansion (Lee et al., 2019; Matena and Raffel, 2022) to measure the importance of the parameters. However, the knowledge conflicts still exist for parameters that have high importance simultaneously.

**Model Routing.** To completely avoid conflicts, another way is to select the most relevant model for each task. This approach typically employs a selection mechanism to activate the most relevant model based on the input task $\tau$ (Yang et al., 2024b) or sequence $x_{1:t}$ at $t$-th token like the Mixture of Experts (MoE) (Tang et al., 2024b). Current methods propose different routing and re-training mechanisms to improve the performance (He et al., 2024a; Wei et al., 2024a; Sukhbaatar et al., 2024b). However, these methods fail to consider merging parameters to find the common knowledge that can be shared across different tasks, and cause large memory and computational costs. We leave detailed discussions about related works in Appendix C).

**Out-of-distribution Data.** In real-world deployment, the test data $x$ may come from other distributions instead of the $p(x|\tau)_{\tau \in \mathcal{T}}$. To this end, we need to consider how to handle OOD data $x$ within merging LLMs. In this work, we mainly consider two OOD cases and tackle them in Section 4.

## 3 UNDERSTANDING CONFLICT BETWEEN LLMS

**Preliminary Experiments.** Before introducing our framework, we first investigate the nature of parameter conflicts in model merging. We conduct preliminary experiments to quantify how conflicts arise and distribute across layers. We finetune Llama-3.2-3B on three datasets and evaluate them on corresponding tasks (experimental details in Section 5). Let $P(\theta, \tau)$ denote

Table 1: Accuracy of finetuning Llama 3.2 3B.

| Model | GSM8K Math | TriviaQA | H.Eval Code | All tasks |
|---|---|---|---|---|
| $\theta$ (Pretrained) | 27.52 | 57.71 | 22.56 | 35.93 |
| $\theta_1$ (Math SFT) | **46.47** | 54.59 | 25.00 | 42.02 |
| $\theta_2$ (QA SFT) | 32.75 | **61.45** | 28.05 | 40.75 |
| $\theta_3$ (Coding SFT) | 33.13 | 57.71 | **40.85** | 43.90 |
| $\phi_{\text{AVG}}$ | 42.61 | 60.99 | 31.30 | 44.97 |
| $\phi_{\text{SEL}}$ | **46.47** | **61.45** | **40.85** | **49.59** |

the performance of model $\theta$ on task $\tau$. We define three baselines: the original pretrained model ($P_{\text{ORI}}$), the simply averaged model ($P_{\text{AVG}}$), and an oracle selection model ($P_{\text{SEL}}$) that selects the optimal finetuned expert $\theta_\tau$ for each task. As shown in Table 1, we observe the performance hierarchy $P_{\text{ORI}} < P_{\text{AVG}} < P_{\text{SEL}}$. This inequality implies two key insights. First, merging knowledge via

averaging ($P_{\text{AVG}}$) improves upon the base model ($P_{\text{ORI}}$). Second, simple averaging fails to fully recover the specialized capabilities of individual experts ($P_{\text{SEL}}$). This gap suggests that indiscriminate averaging dilutes task-specific knowledge due to parameter conflicts.

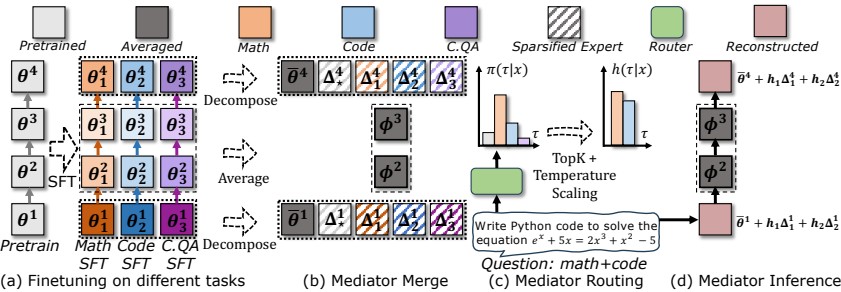

Figure 2: The framework of Mediator.

To close the gap between $P_{\text{AVG}}$ and $P_{\text{SEL}}$, we must identify where conflicts occur. We utilize Task Arithmetic to isolate task-specific updates.

**Definition 3.1** (Task Arithmetic). A task arithmetic on task $\tau$ is the parameter difference between the finetuned LLM $\theta_\tau$ and the pre-trained LLM $\theta$, i.e., $\Delta_\tau = \theta_\tau - \theta$.

**Parameter Denoising.** A critical challenge in measuring conflicts is the presence of stochastic noise. Finetuning is an optimization process where non-essential parameters often undergo random fluctuations (Yadav et al., 2023a; He et al., 2024b). For instance, a parameter might shift by $+10^{-5}$ in Task A and $-10^{-5}$ in Task B purely by chance. Although these shifts are negligible for performance, a raw sign-based metric would incorrectly flag them as a "conflict" ($+$ vs $-$).

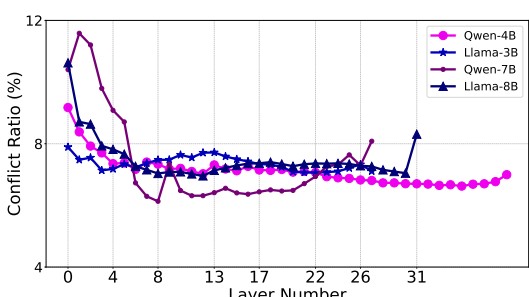

Figure 3: Parameter conflict distribution across different layers of finetuned models (Qwen 2.5 7B).

To prevent such false positives, we denoise the task arithmetic before measurement. We model the update elements of $\Delta_\tau$ as a Gaussian distribution $\mathcal{N}_{\text{UPD}}(\mu_{\text{UPD}}, \sigma^2_{\text{UPD}})$. We identify parameters within the range $(\mu_{\text{UPD}} - \sigma_{\text{UPD}}, \mu_{\text{UPD}} + \sigma_{\text{UPD}})$ as noise and prune them to zero. We then obtain the denoised arithmetic $\hat{\Delta}_\tau$ and the corresponding denoised model $\hat{\theta}_\tau = \theta + \hat{\Delta}_\tau$ (further details in Appendix F.1).

**Measuring Parameter Conflict.** Using the denoised parameters, we measure conflicts using sign consistency (Yadav et al., 2023a). For a layer $l$, given parameters $w_i^l \in \hat{\theta}_{\tau_i}^l$ and $w_j^l \in \hat{\theta}_{\tau_j}^l$, a conflict exists if $sgn(w_i^l w_j^l) = -1$. The conflict ratio $d_l$ is the proportion of conflicting parameters:

$$d_l = \frac{\sum_{i,j} \mathbb{I}(sgn(w_i^l w_j^l) = -1)}{|\theta^l|} \tag{2}$$

**Observation.** Figure 3 illustrates the distribution of $d_l$ across layers. We observe that conflict levels are non-uniform. Specifically, front layers consistently exhibit high conflict ratios, likely due to the divergence in shallow feature extraction for different tasks. In contrast, central layers demonstrate significantly lower conflict, suggesting a region of shared, common knowledge.

It is worth noting that while the final layers often show high conflict (as seen in Llama-3), this is not universal; for example, Qwen-4B exhibits relatively lower conflict in its final layers. This variability in conflict distribution—where the "safe-to-merge" zones differ by model architecture—underscores the necessity for an adaptive merging strategy rather than fixed layer selection.

## 4 THE DESIGN OF MEDIATOR

This section introduces Mediator, a unified framework designed to efficiently merge multiple finetuned LLMs. Our approach begins with **Adaptive Merging** (Section 4.1), which leverages our finding that parameter conflicts are concentrated in the front and last layers. By averaging low-conflict central layers and routing the high-conflict ones, we preserve task-specific capabilities without noise. To address the storage overhead of maintaining these multiple experts, we introduce **Expert Decomposition** (Section 4.2), which utilizes sparsity to decouple experts into lightweight task arithmetics. Finally, we implement **uncertainty-aware Expert Routing** (Section 4.3), a mechanism that dynamically selects or mixes these sparse experts to handle ambiguous or out-of-distribution inputs effectively.

### 4.1 ADAPTIVE LAYER-WISE MODEL AVERAGING AND ROUTING

The core challenge in model merging is that parameter conflicts are not uniformly distributed across the model. As analyzed in Section 3, averaging on high-conflict layers destroys task-specific information, while routing every layer is computationally prohibitive. Mediator solves this via an adaptive merging strategy: we average layers with shared knowledge (low conflict) and route layers with distinct task requirements (high conflict) as shown in Figure 2.

**Dynamic Thresholding.** Instead of manually designating "front" or "back" layers, Mediator dynamically identifies conflict zones. We first calculate the conflict score $d_l$ for each layer $l$ (as defined in Eq. 2). We model the distribution of these layer-wise conflicts as a Gaussian distribution $d_l \sim \mathcal{N}(\mu, \sigma)$. We apply the *Route operation* to layers falling above the adaptive threshold ($\mu + \sigma$) and *Average* those falling below it. Empirically, we observe that layers satisfying $d_l \geq \mu + \sigma$ (high conflict) are primarily located in the front and last layers of the model, corresponding to shallow feature processing and final output formatting as shown in Figure 3. Conversely, central layers typically satisfy $d_l < \mu + \sigma$, indicating they store common knowledge suitable for averaging. Additionally, following findings that attention mechanisms largely store non-specific domain knowledge (Sukhbaatar et al., 2024a), we default to averaging all attention layers.

**Merging Operations.** For layers designated for averaging ($\phi_{\text{AVG}}^l$), we employ TIES-Merging (Yadav et al., 2023a) to compute a unified parameter set. For layers designated for routing ($\phi_{\text{UP}}^l$), we maintain the distinct parameters for each task, which we address in the following subsection.

### 4.2 EXPERT DECOMPOSITION

The adaptive merging strategy creates a new challenge: maintaining $N$ distinct experts for the routed layers ($\phi_{\text{UP}}^l$) requires memory $N \times M_l$, where $M_l$ is the layer size. This linear scaling with the number of tasks ($N$) is inefficient.

To resolve this, we introduce Expert Decomposition. We leverage the observation that while the base weights are dense, the task-specific updates (task arithmetics) are highly sparse and distinguishable

---

**Algorithm 1** Adaptive Merging and Routing with Sparsified Expert Decomposition in Mediator

**Input:** Different finetuned models $\theta_1, \theta_2, \ldots, \theta_{n_\tau}$.
**Output:** The merged layers $\Phi$.

1: Calculate the conflict distribution $\{d_l\}_{l=1,\cdots,|\mathcal{L}|}$;
2: Estimate $\mu, \sigma$ based on $\{d_l\}_{l=1,\cdots,|\mathcal{L}|}$;
3: **for** layer $l = 1, \cdots, |\mathcal{L}|$ **do**
4:     **if** $d_l < \mu + \sigma$ **then**
5:         $\phi_{\text{AVG}}^l = \mathcal{M}(\theta_1^l, \theta_2^l, \ldots, \theta_{n_\tau}^l)$;
6:     **else**
7:         $\bar{\theta}^l = 1/n_\tau \sum_{\tau \in \mathcal{T}} \theta_\tau^l$;
8:         $\Delta_\tau^l = \theta_\tau^l - \bar{\theta}^l, \Delta_\star^l = \theta^l - \bar{\theta}^l$;
9:         $\hat{\Delta}_\tau^l = \text{Denoise}(\Delta_\tau^l)$;
10:        $\phi_{\text{UP}}^l = \left\{ \bar{\theta}^l, \hat{\Delta}_1^l, \hat{\Delta}_2^l, \ldots, \hat{\Delta}_{n_\tau}^l, \hat{\Delta}_\star^l \right\}$;
11:     **end if**
12:     Insert $\phi_{\text{AVG}}^l$ or $\phi_{\text{UP}}^l$ into $\Phi_{\text{AVG}}$ and $\Phi_{\text{UP}}$;
13: **end for**
14: **Return** $\Phi = \{\Phi_{\text{AVG}}, \Phi_{\text{UP}}\}$. =0

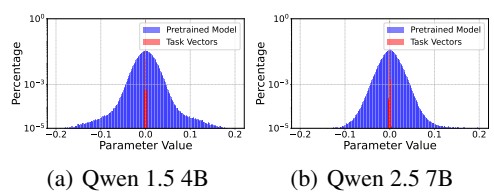

| (a) Qwen 1.5 4B | (b) Qwen 2.5 7B |

Figure 4: Comparing magnitudes of task arithmetic and pretrained model parameters.

---

from noise. As shown in Figure 4, the magnitude of task updates is significantly smaller than the pretrained parameters.

We decompose each routed expert $\theta_\tau$ into a shared base $\theta$ and a sparse task arithmetic $\Delta_\tau$:

$$\theta_\tau = \theta + \Delta_\tau \tag{3}$$

We then sparsify $\Delta_\tau$ into $\hat{\Delta}_\tau$ by pruning parameters below a noise threshold (derived from Section 3). This reduces the storage cost from $N \times M_l$ to roughly $N \times M_l \times c$, where $c$ is the sparsity ratio (typically $< 10\%$).

**Preserving Pretraining Knowledge (OOD-I).** (OOD to $\mathcal{T}$ but in-distribution to pretraining data $p(x|\mathcal{T}^\star)$) Finetuning often induces catastrophic forgetting of the pretraining distribution. To ensure Mediator retains general capabilities for inputs that are Out-Of-Distribution (OOD) to the specific tasks but In-Distribution to the pretraining data ($x \sim p(x|\mathcal{T}^\star)$), we explicitly store the "pretraining arithmetic" $\Delta_\star = \theta - \bar{\theta}$, where $\bar{\theta}$ is the average of the finetuned models. This allows us to revert to the generalist capability when necessary.

### 4.3 EXPERT ROUTING

With the experts decomposed into sparse components $\hat{\Delta}_\tau$, the next challenge is selecting the correct expert during inference like the token-level expert routing in MoE (Sukhbaatar et al., 2024a).

**Task-level vs. Token-level Routing.** Standard MoE models use token-level routing. However, finetuned LLMs are trained on complete sequences $x_{1:T} \sim p_\tau$. Therefore, a subsequence $x_{1:t} \subset x_{1:T}$ might be OOD to the another model $\theta_{\hat{\tau}}$. Switching experts mid-sentence (token-level) fragments the semantic context and degrades performance (theoretical analysis provided in Appendix D). Therefore, Mediator employs *Task-level Routing*, selecting experts once per input sequence. Besides, task-level routing is more system friendly than token-level routing because there is no expert reloading during decoding process.

**Handling Compositional Tasks (OOD-II).** (OOD but close to $\mathcal{T}$ and $\Omega$) A rigid selector fails on inputs $x$ that do not belong strictly to a single training task $\tau$ (e.g., a query requiring both coding and reasoning). To handle such OOD samples, we propose **Uncertainty-based Expert Selection**. We treat the selection as a probabilistic merging problem.

---

**Algorithm 2** Uncertainty based Expert Routing in Mediator

**Input:** The question prompt $x_{1:t}$, router $\pi_\kappa$, $\Phi_{\mathrm{UP}}$, $k$, $\beta$.

**Output:** Reconstructed routing layers $\Phi_{\mathrm{UP},x}$.

1: $\mathcal{T}_k^x = $ Indexes of TopK($\{\pi_\kappa(\tau|x)\}, k$);
2: Calculate and cache $\{h(\tau|x)\}$ (Equation 4);
3: **for** $\phi_{\mathrm{UP}}^l \in \Phi_{\mathrm{UP}}$ **do**
4: $\quad \phi_{\mathrm{UP},x}^l = \bar{\theta}^l + \sum_{\tau \in \mathcal{T}_k^x} h(\tau|x)\hat{\Delta}_{n_\tau}^l$;
5: $\quad$ Insert $\phi_{\mathrm{UP},x}^l$ into $\Phi_{\mathrm{UP},x}$;
6: **end for**
7: **Return** $\Phi_{\mathrm{UP},x}$. =0

---

Specifically, we train a lightweight classifier $\kappa$ to predict the task likelihood. During inference, we apply temperature scaling to the logits to estimate the posterior distribution (uncertainty) $h(\tau|x)$:

$$h(\tau|x) = \frac{e^{\pi_\kappa(\tau|x)/\beta}}{\sum_{\tau \in \mathcal{T} \cup \star} e^{\pi_\kappa(\tau|x)/\beta}} \tag{4}$$

For clear task inputs, this distribution is sharp (selecting a single $\hat{\Delta}_\tau$). For ambiguous or OOD inputs, the distribution flattens, effectively computing a weighted average of relevant experts. The final parameters for the routed layer are constructed dynamically as $\theta_{\mathrm{routed}} = \theta_{\mathrm{base}} + \sum_\tau h(\tau|x)\hat{\Delta}_\tau$. This process is summarized in Algorithm 2.

### 4.4 SYSTEM-LEVEL OPTIMIZATION

The dynamic construction of $\theta_{\mathrm{routed}}$ introduces latency due to memory loading. We implement two system-level optimizations to mitigate this overhead:

**Asynchronous Prefetching.** The bottleneck lies in loading sparse experts from CPU/disk to GPU. We implement asynchronous prefetching: while the GPU computes layer $l$, the CPU prepares and

transfers the sparse experts for layer $l + 1$ (i.e., Algorithm 2). This effectively masks the loading latency behind the computation time.

**Fused Sparse Integration.** To accelerate the summation $\theta + \sum h\hat{\Delta}$, we utilize a custom CUDA kernel based on atomicAdd. This allows us to parallelize the addition of sparse matrices directly into the dense backbone without materializing intermediate dense tensors.

For non-sparse components stored on disk, we utilize ZipNN (Hershcovitch et al., 2024) for rapid decompression. As detailed in Appendix G, these optimizations result in an inference latency only $0.2s \sim 0.4s$ higher than a standard single model, despite the dynamic routing capabilities.

# 5 EXPERIMENTS

**Models and Evaluation Tasks.** We conduct comprehensive experiments on cutting-edge LLMs including Qwen-1.5-4B, Qwen-2.5-7B (Yang et al., 2024c), LLaMA-3.1-3B, and LLaMA-3.2-8B (Dubey et al., 2024). We select different evaluation tasks to effectively demonstrate model capability in resolving parameter conflicts during model merging, including GSM8K of mathematical question-answering (Cobbe et al., 2021), TriviaQA (Joshi et al., 2017) of a large-scale Wikipedia-based question answering dataset, HumanEval (Chen et al., 2021) of Python programming tasks, WinoGrande (Sakaguchi et al., 2019) of logical reasoning, MMLU (Hendrycks et al., 2021) of vertical domain knowledge (as OOD to the finetuned models).

**Finetuning Settings.** The finetuning datasets are constructed by augmenting some publicly datasets (task related but without overlap) with GPT-4o (Gilardi et al., 2023) and Chain-of-Thoughts (Wei et al., 2022). For each finetuning process, we use at least 180K training samples to ensure sufficient performance improvement on the corresponding task, which helps validate the effectiveness of our experiments (Details of constructing finetuning datasets in Appendix H and hyperparameters in Appendix E). *To the best of our knowledge, this is the first LLM merging study with CoT enhanced finetuning and evaluated with generative tasks.*

Table 2: Comparing performance of model merging methods on Llama 3.1 3B and Llama-3.2 8B.

| Model | Llama 3.1 3B | | | | | | Llama-3.2 8B | | | | | |
|---|---|---|---|---|---|---|---|---|---|---|---|---|
| Alg./Tasks | GSM. | TrA. | Wino. | H.Eval | MMLU | AVG. | GSM. | TrA. | Wino. | H.Eval | MMLU | AVG. |
| Base | 27.52 | 57.71 | **69.69** | 22.56 | **54.08** | 46.31 | 56.33 | 72.39 | 73.64 | 27.44 | **67.99** | 59.56 |
| Math | **46.47** | 54.59 | 69.06 | 25.00 | 52.73 | 49.57 | **77.18** | 73.99 | 74.98 | 20.12 | 62.10 | 61.67 |
| QA | 32.75 | **61.45** | 69.69 | 28.05 | 54.17 | 49.22 | 69.60 | 74.14 | **75.45** | 31.71 | 62.21 | 62.43 |
| Code | 33.13 | 57.71 | 68.59 | **40.85** | 53.09 | **50.67** | 61.41 | 73.94 | 74.59 | **62.80** | 62.73 | **67.09** |
| All data | 44.12 | 47.74 | 69.21 | 34.76 | 53.75 | 49.92 | 70.89 | 69.77 | 75.06 | 48.17 | 62.94 | 65.37 |
| TIES | 42.61 | 60.99 | 71.11 | 31.30 | 54.32 | 51.27 | 76.04 | 76.78 | 74.19 | 53.05 | 62.36 | 68.48 |
| PCB | 46.02 | 60.39 | 71.27 | 29.88 | 54.21 | 52.35 | 76.04 | 76.89 | 74.35 | 53.66 | 62.42 | 68.67 |
| Twin | 39.04 | 52.45 | 69.27 | 29.94 | 53.91 | 48.11 | 76.80 | 72.71 | 74.49 | 59.14 | 64.43 | 69.51 |
| BTX | 45.19 | **62.05** | 71.87 | 28.05 | 54.44 | 52.33 | 76.72 | 73.99 | 75.22 | 60.98 | 65.68 | 70.52 |
| **Mediator** | **46.47** | 61.02 | **72.03** | **40.42** | **54.91** | **54.97** | 76.95 | 76.70 | 75.69 | 62.80 | 67.87 | **71.80** |

Table 3: Comparing performance of model merging methods on Qwen 1.5 4B and Qwen 2.5 7B.

| Models | Qwen 1.5 4B | | | | | | Qwen 2.5 7B | | | | | |
|---|---|---|---|---|---|---|---|---|---|---|---|---|
| Alg./Tasks | GSM. | TrA. | Wino. | H.Eval | MMLU | AVG. | GSM. | TrA. | Wino. | H.Eval | MMLU | AVG. |
| base | 47.16 | 44.54 | 56.75 | **41.46** | 54.45 | 48.87 | 83.41 | 51.67 | 67.68 | 67.68 | 67.70 | **67.63** |
| Math | 51.00 | 46.95 | 54.62 | 26.83 | 53.54 | 46.79 | **85.14** | 51.67 | 65.75 | 61.59 | 67.27 | 66.27 |
| Code | 43.29 | 46.39 | 54.14 | 43.29 | **54.82** | 48.39 | 52.31 | 49.47 | 64.64 | **71.95** | **72.30** | 62.13 |
| QA | 45.56 | **48.02** | 57.93 | 39.02 | 52.32 | 48.57 | 84.62 | **55.58** | 62.83 | 43.29 | 71.51 | 63.57 |
| all-sft | 48.52 | 47.73 | 55.88 | 39.14 | 53.93 | **49.04** | 64.90 | 52.98 | 69.30 | 65.85 | 69.66 | 64.59 |
| TIES | 47.76 | 46.59 | 54.14 | 44.51 | 54.58 | 49.5 | 84.76 | 54.46 | 66.46 | 65.85 | 71.55 | 68.62 |
| PCB-merging | 47.83 | 47.60 | 56.75 | 43.90 | 54.58 | 49.93 | 73.46 | 53.90 | 69.53 | 60.98 | 71.41 | 65.86 |
| Twin-merging | 47.99 | 44.63 | 57.54 | 40.85 | 52.98 | 48.80 | 83.46 | 54.64 | 66.37 | 70.56 | 66.37 | 68.91 |
| BTX | 48.44 | 46.94 | 57.77 | 42.68 | 53.88 | 49.94 | 84.46 | 55.89 | 67.72 | 67.68 | 72.30 | 69.61 |
| Mediator | **50.94** | **48.20** | **57.85** | **45.12** | **54.87** | **51.40** | **85.14** | **56.06** | **69.30** | **71.95** | **72.56** | **71.00** |

**Baselines.** We compare pretrained, finetuned models, and the state-of-the-art static and dynamic merging methods with Mediator. The *static merging methods* include TIES (Yadav et al., 2023b) and PCB-merging (Du et al., 2024) achieve the best performance in weighted average method and do not require calibration data, and also partly consider OOD evaluation tasks. The dynamic merging methods include BTX (Sukhbaatar et al., 2024b) with token-level routing and the twin-merge

with task-level routing and SVD decomposition (Lu et al., 2024b) (Details of hyperparameters and optimization of these baselines in Appendix E).

## 5.1 MAIN RESULTS

**Fine-grained Comparison on All Tasks.** Table 2 and 3 show the fine-grained performance on each tasks and their overall averaged one of different methods and algorithms. In most of time, the finetuned LLM can achieve the best performance across all single and merged models on its specialized domain, like Math finetuned models on GSM8K and Code finetuned models on HumanEval. While merged LLMs can generally outperform single models on the averaged performance, their specialized domain performance is weaker. However, Mediator can catch up the domain performance of specialized models, and almost always outperform other merged models. Also, the overall performance on all tasks of Mediator is consistently better than other methods. Expert routing methods includes BTX and Mediator generally improve performance. This aligns with findings in TIES (Yadav et al., 2023b) and Twin-merging (Lu et al., 2024a). As model scale increases, the improvement of all merging algorithms decrease, which may be attributed to enhanced comprehensive capabilities of individual finetuned models.

**Overall Comparison.** As shown in Table 4, the advantages of PCB over TIES become less pronounced at larger model scales, and even shows performance degradation on Qwen-7B, which demonstrates PCB's instability. Dynamic routing approaches include BTX and Mediator show stable performance improvements. Our method demonstrates consistent improvements across different models.

Table 4: Overall Performance on all tasks.

| Model/Algo. | Qwen-4B | Llama-3B | Qwen-7B | Llama-8B |
|---|---|---|---|---|
| base model | 48.87 | 46.31 | 67.63 | 59.56 |
| all data sft | 49.04 | 49.92 | 64.59 | 65.37 |
| TIES | 49.50 | 51.27 | 68.62 | 68.48 |
| Twin | 48.80 | 48.11 | 68.91 | 69.51 |
| PCB | 49.93 | 52.35 | 65.86 | 68.67 |
| BTX | 49.94 | 52.33 | 69.61 | 70.52 |
| **Mediator** | **51.40** (↑2.9%) | **54.97**(↑5.0%) | **71.00** (↑2.0%) | **71.80** (↑1.8%) |

**Post-Training Time After Merging.** As many model merging methods like Twin, PCB and BTX require post-training, it is critical to compare the extra training time. Table 5 shows the post-training time of different methods. PCB merging require weight exploration thus leads to higher time. The BTX with token-level routing needs to completely train the layer-wise routers for each token, thus,

Table 5: Post Training Time (Hours).

| Model/Algo | LLaMA-3B | LLaMA-8B | Qwen-4B | Qwen-7B |
|---|---|---|---|---|
| Twin | 1.33 | 1.87 | 1.60 | 1.80 |
| PCB | 3.42 | 5.75 | 5.73 | 5.80 |
| BTX | 7.55 | 12.52 | 8.83 | 12.18 |
| Mediator | 1.35 | 2.03 | 1.57 | 1.78 |

the post-training time of them is significantly high. In contrast, for task-level routing approaches like Twin-merging and Mediator, taking the lowest time.

**Inference Time.** Table 6 shows the inference time of different methods. As the token-level routing methods need to load and compute the layer-wise routers for each token, the inference time of them is significantly higher (more than 2x) than our method. We have detailed our inference acceleration process in Section 4.4 and Appendix G.

Table 6: Inference time per sample (seconds).

| Model/Algo | LlaMA-3B 32 layers | LlaMA-8B 32 layers | Qwen-4B 40 layers | Qwen-7B 28 layers |
|---|---|---|---|---|
| Base model | 1.452 | 3.600 | 3.112 | 3.057 |
| Twin | 1.725 | 4.151 | 3.792 | 3.648 |
| BTX | 3.237 | 8.68 | 7.082 | 7.153 |
| Mediator | 1.609 | 4.053 | 3.674 | 3.489 |

Table 7: Comparing memory costs.

| Model/Algo | LLaMA-3B | LLaMA-8B | Qwen-4B | Qwen-7B |
|---|---|---|---|---|
| Base model | 9G | 33G | 11G | 31G |
| Twin-merging | 10G | 35G | 13G | 32G |
| BTX (MOE) | 37G | 80G | 40G | 78G |
| Mediator | 10G | 35G | 13G | 33G |

**Memory Cost.** Table 7 shows the memory costs of different methods. Our method significantly reduces the memory costs compared to saving all finetuned models because there is only one router for all experts, and the sparsified experts saving, and the layer-wise merging strategy (Details of formally comparing memory costs of merging methods in Appendix C).

## 5.2 ABLATION STUDIES

**Scalability of Finetuned Models.** To verify the scalability of Mediator, we finetune another 4 LLMs according to the following 4 extra evaluation tasks including: (1) *Instruction Following* with IFEval. (Zhou et al., 2023) which assess models ability to accurately interpret and execute natural language instructions; (2-4) *Medicine, College Economics and Law* from CEval which assess

knowledge and capabilities across various academic and professional domains (Huang et al., 2023a). We utilize four accordingly domain datasets for finetuning including Magpie (Xu et al., 2024), IndustryInstruction (Ind), DISC-Med (Bao et al., 2023), DISC-Law (Yue et al., 2023a) without overlap with IFEval and CEval (Details in Appendix H).

Table 8 demonstrates several key findings: 1) Static merging methods like PCB and TIES show diminished performance improvements after task expansion, performing even worse than instruction-following finetuned models in overall scores. The similar performance between PCB merging and TIES aligns with findings from the TIES paper, which noted degraded model capabilities when merging more than three tasks; 2) Dynamic

Table 8: Performance of scaling up finetuned models.

| Alg./Tasks | GSM. | TrA. | Wino. | H.Eval | MMLU | I.Eval | C.Eco | C.Med | C.law | AVG. |
|---|---|---|---|---|---|---|---|---|---|---|
| Base | 47.16 | 44.54 | 56.75 | 41.46 | 54.45 | 30.70 | 49.09 | 55.10 | 41.67 | 46.77 |
| Math | **50.95** | 46.95 | 54.62 | 26.83 | 53.54 | 34.05 | 45.45 | 55.10 | 45.83 | 45.92 |
| QA | 45.56 | **48.02** | **57.93** | 39.02 | 52.32 | 31.65 | 43.64 | 59.18 | 45.83 | 47.02 |
| code | 43.29 | 46.39 | 54.14 | **43.29** | 54.82 | 31.65 | 43.64 | 59.18 | 45.83 | 46.91 |
| Instruct. | 47.54 | 40.96 | 55.09 | 37.80 | **54.88** | **38.37** | 52.73 | 59.18 | 50.00 | 48.51 |
| Economy | 45.56 | 46.24 | 57.93 | 28.86 | 54.21 | 32.13 | **56.36** | 55.10 | 45.83 | 46.78 |
| medicine | 39.12 | 44.50 | 56.67 | 1.83 | 54.63 | 28.30 | 50.09 | **61.22** | 41.66 | 42.00 |
| Law | 40.64 | 46.64 | 56.59 | 0.61 | 54.61 | 27.94 | 41.82 | 57.14 | **58.33** | 47.15 |
| All Data | 43.75 | 46.25 | 56.43 | 40.85 | 54.60 | 35.37 | 49.09 | 55.10 | 41.67 | 46.90 |
| TIES | 47.38 | 47.19 | 55.80 | 36.59 | 55.38 | 34.17 | 40.00 | 57.14 | 50.00 | 47.07 |
| PCB | 47.38 | 47.19 | 55.80 | 36.59 | 55.45 | 34.29 | 41.82 | 57.14 | 50.80 | 47.38 |
| Twin | 47.91 | 44.78 | 57.54 | 40.85 | 53.01 | 37.53 | 53.32 | 59.56 | 50.00 | 49.38 |
| BTX | 48.44 | 46.94 | 57.85 | 42.68 | 54.93 | 36.93 | 54.40 | 60.36 | 58.33 | 51.18 |
| **Mediator** | 50.64 | **48.04** | **57.93** | **44.51** | **55.12** | **38.50** | **56.01** | 61.17 | **58.33** | **52.25** |

merging approaches like BTX, Twin and Mediator maintain relatively stable performance after task expansion; 3) Mediator consistently outperforms BTX by a margin of 2.09% and achieves the best scores across all individual tasks, showing its good scalability.

**Layer-wise Merging.** Layers with low conflicts are averaged thus reducing $n_\tau \times$ memory occupation. Table 9 averaging or not on Qwen-1.5 with 4 and 8 experts show almost no performance loss of the parameter averaging. Given the significant compression ratios achieved (3.5x for 4 experts, 7x for 8 experts), this minor performance trade-off is acceptable.

Table 9: Model Performance w/o layer-wise merging.

| Model scale | with averaging | w/o averaging | perf. gap |
|---|---|---|---|
| Qwen-1.5 | 51.40 | 51.43 | -0.06% |
| Qwen-2.5 | 71.00 | 71.29 | -0.27% |

**Comparison between Task-level routing and Token-level routing.** Fig. 7 in Appendix demonstrates the expert selection probabilities of the BTX model across different tasks (MMLU, math, coding, and QA). The results indicate that 1) bottom and upper layers show obvious task preference; 2) middle layers suggesting some shared logical processing across tasks in these layers, which explains why averaging middle layers that have less conflicts in Mediator results in small performance loss.

**Different Temperatures.** In the routing process, the hyperparameter temperature $\beta$ is a key factor. Table 10 shows performance change of Mediator with different temperatures. Results show that only around 2.2% score variation, the $\beta = 1.5$ achieves the highest performance, $\beta = 0.1$ al-

Table 10: Performance under Different Temperature $\beta$.

| Temperature | 0.1 | 0.5 | 1.0 | 1.25 | 1.50 | 1.75 | 2.0 |
|---|---|---|---|---|---|---|---|
| Mediator | 50.31 | 50.35 | 50.64 | 51.02 | 51.40 | 50.84 | 50.92 |

most equals to the Top-1 routing, results in the lowest performance.

# 6 CONCLUSION

We propose Mediator, a framework for merging LLMs that addresses parameter conflicts through adaptive layer-wise strategies, which average low-conflict layers and routing high-conflict ones via task-specific experts. This preserves individual model strengths while integrating shared knowledge, improving performance and reducing system costs.

By decomposing experts into a dense core and sparse components, Mediator minimizes storage without sacrificing efficacy. Dynamic expert selection via task uncertainty enhances adaptability across diverse inputs. Experiments on LLaMA and Qwen demonstrate significant performance gains over existing methods, with CoT enhanced datasets further enhancing reasoning capabilities.

Mediator advances efficient LLM merging methods, balancing resource constraints with practical versatility. Future work should explore theoretical foundations of parameter conflicts, large-scale deployment optimizations, and faster expert loading mechanisms.

ETHICS STATEMENT

We declare no conflicts of interest that could inappropriately influence our work. All experiments were conducted using publicly available resources. Our study does not involve human subjects, data collection from individuals, or experiments on protected groups. The models and basic datasets used in this work are publicly available and widely used in the research community. We have made efforts to ensure our experimental design and reporting of results are fair, unbiased, and do not misrepresent the capabilities or limitations of the methods presented.

REPRODUCIBILITY STATEMENT

For openness of LLM research, we declare our code and the CoT enhanced crafted finetuning datsets will be made available to ensure reproducibility. We will provide detailed documents of code implemnetation. And we have provided the details of all hyper-parameters of implementing Mediator and optimizing baselines.

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

APPENDIX

# A  THE USE OF LARGE LANGUAGE MODELS

We used LLMs solely for grammar and wording improvements. It did not generate ideas, analyses, or results. No additional or undisclosed LLM use occurred.

# B  BROADER IMPACT

**Societal Impacts.** Our approach demonstrates significant effectiveness by enabling the deployment of merging 7B x 4 LLMs with only 24GB VRAM. Compared to ensemble learning with these models, our method not only maintains better accuracy but also requires significantly less computational resources and demonstrates superior performance. This breakthrough in resource efficiency makes advanced language models more accessible and cost-effective.

**Potential Applications.** The technology may have significant potential across specialized vertical domains. Considering that many vertical domains, personalized LLM agents (Li et al., 2024b), LLM applications like roleplay chatting (Chan et al., 2024; Yu et al., 2024a) and professional domain-specific writing (Gómez-Rodríguez and Williams, 2023), an LLM service provider may need to simultaneously deploy different finetuned LLMs. Our technology enables efficient and effective serving multiple popular LLM applications, and merging knowledge from different LLMs together.

# C  MORE RELATED WORKS

We introduce more related works about model merging and routing in this section. Current common methods include: (1) **Averaging based merging.** This direction combines multiple models into a single model while preserving their capabilities with minimal or no additional training; (2) **Routing based merging.** It considers to route inputs to specialized expert modules like mixture-of-experts (MoE) approaches but completely different.

Table 11 provides an overview of comparing different model merging methods and our framework. Given $n_\tau$ different finetuned models, averaging based methods do not completely address the parameter conflicts, thus having higher parameter conflicts than routing based merging. The weighted averaging requires calibration data to compute the importance metrics. The token-level routing requires routing for each layer, thus having totally $n_\mathcal{L}$ routers. Our framework Mediator exploits layer-wise characteristics to both reduce parameter conflict and improve common knowledge fusion. And Mediator utilizes compression to further reduce the memory costs. To the best of our knowledge, the most of previous model merging works focus on experiments on traditional CV and NLP tasks (Matena and Raffel, 2022), while Mediator conducts experiments on modern LLMs and real-world experiments.

Table 12 provides the system performance comparison. Because token-level routing like MoE requires to route each token towards different (possibly) experts, its inference cost is significantly large as more than $T \times n_\mathcal{L}$ times than task-level routing, where $T$ is the sequence length. Besides, the token-level routing requires more than $n_\mathcal{L}$ times routers in memory costs. With the layer-wise adaptive averaging and Routing, Mediator significantly reduce the memory costs of from $M_\theta \times n_\tau$ to $M_\theta \times (c_{\text{avg}} + c_{\text{route}} \times n_\tau \times c)$ [1]. The experimental memory reduction and the system performance comparisons are shown in the Section 5.1. We also provide system optimization to accelerate the inference during the deployment of Mediator in Section 4.4 and Appendix H. And the hyper-parameters $c_{\text{avg}}$ and $c_{\text{route}}$ are adaptively decided by the parameter conflict estimation.

Besideds, we also review some highly related works include following directions that are closely related to our framework. Insights from these directions have provided valuable guidance for our framework.

1. **Layer-wise training dynamics and optimization.** This direction discusses the layer-wise training dynamics to help shed some light on the paramter conflicts and the layer-wise adaptivity.

---

[1]Normally, each transformer layer occupies the same memory.

Table 11: Demystifying different merging methods. The $n_\tau$ represents the number of finetuning tasks, $n_\mathcal{L}$ the number of layers in the model.

| Method Type | Parameter Conflict Level | Merging Common Knowledge | Require Calibration Data | Routing Type | Considering Layer-wise Characteristics | Considering Compression | Considering OOD Samples | Experimental Scenario |
|---|---|---|---|---|---|---|---|---|
| Basic Averaging | High | ✗ | ✗ | NA | ✗ | ✗ | ✗ | Traditional CV, NLP |
| Weighted Averaging | Middle | ✓ | ✓ | NA | ✗ | ✗ | ✗ | Traditional CV, NLP |
| Subspace Averaging | Middle | ✓ | ✗ | NA | ✗ | ✗ | ✗ | Traditional CV, NLP |
| Token-Level Routing | **Low** | ✗ | ✗ | Token-level | ✗ | ✗ | ✗ | Traditional CV, NLP |
| Task-Level Routing | **Low** | ✗ | ✗ | Task-level | ✗ | ✗ | ✗ | Traditional CV, NLP |
| Mediator | **Low** | ✓ | ✗ | Task-level | ✓ | ✓ | ✓ | Generative LLMs |

Table 12: Demystifying different merging methods in system performance costs. Considering the memory costs of the base model and one router are $M_\theta$ and $M_h$, each layer occupies the same memory $M_l$, compression ratio $c$, the ratio of selected layers for averaging is $c_{\text{avg}}$, for routing is $c_{\text{route}}$, $FP_\theta$ and $BP_\theta$ are the forward time and backward time of the model. $FP_h$ and $BP_h$ are the forward time and backward time of the router.

| Method Type | Requiring # of routers | Costs of Trainig Routers | Memory Costs After Merging | Inference Cost |
|---|---|---|---|---|
| Basic Averaging | NA | NA | $M_\theta$ | $FP_\theta$ |
| Weighted Averaging | NA | NA | $M_\theta$ | $FP_\theta$ |
| Subspace Averaging | NA | NA | $M_\theta$ | $FP_\theta$ |
| Token-Level Routing | $n_\mathcal{L} \times n_\tau$ | High | $M_\theta \times n_\tau + M_h \times n_\mathcal{L}$ | $FP_\theta + FP_h \times T \times n_\mathcal{L} \times n_\tau$ |
| Task-Level Routing | $n_\tau$ | Middle | $M_\theta \times n_\tau + M_h$ | $FP_\theta + FP_h \times n_\tau$ |
| Mediator | $n_\tau$ | Low | $M_\theta \times (c_{\text{avg}} + c_{\text{route}} \times n_\tau \times c) + M_h$ | $FP_\theta + FP_h \times n_\tau$ |

2. **Bayesian deep learning.** This direction reviews some works of the Bayesian deep learning, discussing the uncertainty and Bayesian model averaging.

3. **OOD Detection & Generalization.** This direction reviews some works of the OOD Detection and Generalization, shedding light on deployment of the model merging on the out-of-distribution data.

4. **Model compression.** This direction shortly review some works about the model compression, in which many methods can be directly applied into our framework to further reduce the memory costs. Note that in our paper we propose a general framework instead of a new model compression method. Different model compression methods can be combined into our framework.

5. **Data Synthesis.** This direction reviews some works about how to generate new synthetic data to improve the model merging performance. In our framework, we exploit the CoT to generate new synthetic data to improve the finetuning performance on downstream tasks, which is a real-world downstream task instead of traditional model fine-tuning using the in-domain training and testing data.

C.1 AVERAGING-BASED MODEL MERGING

Model merging, also known as model fusion, combines the parameters of multiple separate models with different capabilities to create a universal model. In this paper, we temporarily focus on models that have the same architecture but different parameters that are finetuned on different downstream tasks.

**Basic Averaging.** The traditional approach to merge different trained or finetuned models is to evenly average the parameters of different models (Utans, 1996; Shoemake, 1985). This process does not require access to the original training data and allows for enhanced performance without the need for expensive computation. However, the performance of these simply weight averaging is generally unsatisfactory.

Some related directions of model averaging also include Federated learning (FL) (McMahan et al., 2017; Tang et al., 2020). In FL, the model averaging is performed on the server side to reduce the communication costs after the local training. Many methods have been proposed to stable and smooth the model averaging process (Wang et al., 2020a; Jhunjhunwala et al., 2024a; Yurochkin et al., 2019a; Singh and Jaggi, 2020; Wang et al., 2020b; Tang et al., 2024c; 2022) to enhance the averaging performance. Different from the multi-rounds FL, the model merging is performed in a single round, which is more similar to the one-shot FL (Guha et al., 2019; Tang et al., 2024d).

**Weighted Averaging.** Rethinking the cause of the poor performance of the basic averaging method, many works propose to use the weighted averaging method to improve the merging performance.

Intuitively, different model parameters have different importance on downstream tasks. Such a heterogeneity of the parameter importance motivates other research directions including model sparsification (Sun et al., 2024; Dong et al.; 2024; Tang et al., 2020), continual learning (Robins, 1995; Kirkpatrick et al., 2017; Zhu et al., 2024; Marczak et al., 2024) and FL (Jhunjhunwala et al., 2024a; Yurochkin et al., 2019a; Singh and Jaggi, 2020). Thus, to avoid the important parameters being overwhelmed by the unimportant parameters, during averaging, we can assign large weights to those important parameters. To this end, the importance measurement is crucial. Many works propose to use *fisrt or second orders of Taylor expansion* to measure the importance of the parameters (Lee et al., 2019; Jhunjhunwala et al., 2023; Qu et al., 2022). Some works employ *local linearization and task vectors* to measure the importance of the parameters (Zhou et al., 2024). The *fisher information* also a kind of importance measurement (Matena and Raffel, 2022; Jhunjhunwala et al., 2024b; Thennal et al., 2024; Jhunjhunwala et al., 2024a; Thennal et al., 2024; Daheim et al., 2024).

While these importance measurement methods can improve the merging performance than the basic averaging method, they still face some typical challenges.

1. *Require Calibration Dataset.* The importance measurement is based on the calibration dataset. In the LLM era, the pretrained dataset is significantly large, it is difficult to collect the complete pretrained dataset and measure the importance of the parameters on it.

2. *Computation Costs.* Because that the importance measurement is based on the calibration dataset, the computation costs is almost similar to conduct the complete forward process of the different models. In traditional small models, such a computaton cost is acceptable. However, in the LLM era, the model size is significantly large, such a computation cost is unbearable.

3. *Unaddressed Parameter Conflicts.* While methods in these importance based weighted averaging methods can improve the merging performance, they still face the parameter conflicts between different models. Because of the highly non-convex structure of the LLMs, it is difficult to find a optimal merging method based on averaging the parameters of different models.

**Subspace Averaging.** Considering that the neural networks are over-parameterized, removing most of the parameters from the model barely affects its accuracy (He and Xiao, 2023; Choudhary et al., 2020). Besides, during the training or finetuning, some parameters might be optimized towards a random direction which has small impact on the model performance (Yadav et al., 2023b). Thus, works propose to firstly process different models in a subspace manner. Then, the parameter conflicts can be mitigated by the subspace averaging methods (Deep et al., 2024; He et al., 2024b).

DARE (Drop and Rescale) (Yu et al., 2024b;c) introduces a parameter pruning and rescaling strategy that significantly reduces the number of parameters in SFT models while preserving their performance, thereby serving as an effective preprocessing step for model merging. Similarly, Model Breadcrumbs (Davari and Belilovsky, 2023) enhances sparsification by eliminating both low-magnitude parameters and outlier parameters with exceptionally high weights, thereby reducing noise and improving the generalization of hyperparameters during model merging.

TALL-masks (Wang et al., 2024) creates task-specific mask matrices based on predefined thresholds tailored to individual models, while Model Tailor (Zhu et al., 2024) further refines this approach by masking parameters according to their sensitivity to loss changes and deviations from pretrained values. APL (Kong et al., 2024) advances parameter importance estimation through causal interventions, providing a robust metric for selective parameter retention.

EMR-Merging (Huang et al., 2024) departs from traditional model merging by maintaining a shared model across multiple tasks alongside sparse task-specific models, where each shared parameter is determined by the maximum value among corresponding parameters from all models. Concrete (Tang et al., 2023) further innovates by framing mask construction and model merging as a learnable bi-level optimization problem, with the outer level optimizing the mask matrix and the inner level performing model merging and optimization utilizing unlabeled test samples.

Task Arithmetic (Ilharco et al., 2022) exploits parameter-space arithmetic operations, treating model parameters as vectors and employing addition and subtraction to synthesize new model capabilities. However, many of these approaches, including DARE and Task Arithmetic, heavily rely on hyperparameters for parameter fusion, which can negatively impact the performance of model merging. Additionally, as highlighted in studies such as TIES (Yadav et al., 2023a) and Crisostomi (Crisostomi

et al., 2024), model merging often encounters parameter conflicts that degrade performance when integrating multiple models.

Addressing these challenges, TIES (Trim, Elect, and Disjoint Merge) (Yadav et al., 2023a) implements a comprehensive approach by trimming parameters based on magnitude, selecting relevant weights, and disjointly merging weights using outcomes from task arithmetic operations. This methodology mitigates parameter conflicts and enhances the overall performance of the merged model, positioning TIES as a robust solution in the domain of model merging.

## C.2 ROUTING-BASED MODEL MERGING.

Average-based methods primarily aim to enhance the averaging process of client models. However, the inherently non-linear architecture of deep neural networks complicates the derivation of a globally comparable model through simple averaging.

The basic, weighted-based, and subspace-based merging methods are *static* merging techniques. This implies that the merged model remains consistent across all samples or tasks. Given the variability among input samples and tasks, the model's performance can fluctuate when processing diverse inputs. To this end, certain studies advocate for the *dynamic* merging of models (or subsets of layers) tailored to specific samples or tasks (Li et al., 2024c; Muqeeth et al., 2024; Tang et al., 2024b; Lu et al., 2024a; Kang et al., 2024; Tang et al., 2024d; Shen et al., 2024) during the inference phase.

For each input instance, SMEAR (Muqeeth et al., 2024) initially computes a weighted average of the parameters from each expert by leveraging the distribution of router inputs to the expert modules. This approach maintains a computational cost comparable to that of a single expert. Similarly, Twin-Merging (Lu et al., 2024a) adaptively integrates task-shared and task-specific knowledge based on routing mechanisms during inference. In the same vein, Weight-Ensembling MoE (Tang et al., 2024b) introduces a dynamic merging Transformer architecture. This method identifies that the parameters of the linear layer in the fine-tuned model undergo more significant changes compared to the nonlinear layers, which adversely affects merging performance. Consequently, Weight-Ensembling MoE employs a standard weighted average for all modules except the linear layer, which is dynamically weighted and merged based on the routing network (utilizing sample features as input and merging coefficients as output) during inference. PWE MoE (Tang et al., 2024e) extends Weight-Ensembling MoE to a multi-objective optimization framework, incorporating the preference vector as an input for routing.

*AdaMerging* (Yang et al., 2024b) adaptively learns merging coefficients in a task-aware or layer-wise manner, offering an automated and unsupervised approach to task arithmetic. While this method significantly enhances performance, it incurs high computational costs. *PCB Merge* (Du et al., 2024) introduces a parameter importance detection mechanism that accounts for parameter conflicts and employs heuristic algorithms to explore model fusion parameters, thereby achieving superior results. *TwinMerge* (Lu et al., 2024b) utilizes LoRA or SVD techniques in conjunction with supervised training for parameter fusion, resulting in improved performance.

Nevertheless, these methods encounter inherent limitations. Both AdaMerging and PCB Merge utilize static fusion approaches, which can lead to performance degradation when the actual sample distribution varies during runtime. Meanwhile, TwinMerge performs parameter fusion at the task level; however, the application of LoRA and SVD matrix decomposition markedly reduces model accuracy and introduces substantial online computational overhead. Besides, the code implementation of the TwinMerge actually exploits the LoRA finetuning to replace SVD decomposition. Using SVD decomposition in compressing model parameters leads to disturbed LLMs and significantly degraded model performance.

*Mixture-of-Experts (MoE)* (Jacobs et al., 1991; Jordan and Jacobs, 1994) is a foundational model concatenation and routing strategy comprising multiple expert networks and a router that dynamically selects relevant experts based on the input. This methodology has been extensively adopted in large language models, offering significant reductions in computational costs while preserving model performance. Recent studies, particularly sparse gated MoE (Shazeer et al., 2017) in transformer-based large language models (Lepikhin et al., 2020), have concentrated on maintaining load balancing among experts during training (Zhou et al., 2022; Jiang et al., 2024b), reducing training costs (Dai

et al., 2024), and mitigating performance degradation due to uncoordinated expert training (Chi et al., 2022).

*Upcycling Methods* have been developed to alleviate the high computational demands of training MoE models from scratch by initializing experts from existing dense models. These methods encompass copying existing dense models as experts (He et al., 2024a; Wei et al., 2024a), introducing noise to the MLP layers of dense models to create experts (Noise upcycling) (Chen et al., 2024), and drop upcycling (Anonymous, 2024), which combines parameter dropout with expert copying during training to enhance model robustness, reduce overfitting, and improve performance.

*Branch-Train-Merge (BTM)* (Li et al., 2022) and *Branch-Train-Mix (BTX)* (Sukhbaatar et al., 2024b) are methodologies aimed at further optimizing model training efficiency. These approaches employ different SFT-trained dense models derived from the same base LLM as MoE experts. The experts are interconnected via a router without necessitating additional training, while non-expert components are amalgamated through model merging techniques such as parameter averaging. Only the router undergoes training, thereby substantially reducing overall training costs. Although these methods achieve lower training expenses and marginally outperform traditional model merging approaches, our research indicates that token-level routing can partially degrade model performance. Additionally, maintaining all experts in GPU memory leads to significant parameter redundancy and escalates inference costs, which motivates our ongoing research endeavors.

However, the token-level routing methods are not suitable for model merging. We have provided detailed discussions in the main text Section 4.1 and Appendix D. The token-level routing methods after merging normally require re-training based on all training datasets to obtian a better token-level router, which significantly increases the computational costs, which is discussed in the main text Section 4.4 and Appendix G.

**LoRA based Routing.** Routing samples to different LoRA experts is a promising direction to dynamically route the input to different LoRA experts. This direction includes the *LoraHub* (Huang et al., 2023b) and *sLora* (Babakniya et al., 2023), which explore serving multiple LoRA adapters through techniques like unified paging and tensor parallelism. However, these methods do not consider the better dynamic expert merging method to further improve the model merging performance. In real-world applications, the input distribution is dynamic and the input samples are diverse, which motivates our ongoing research endeavors. Besides, their reliance on LoRA matrix decomposition significantly degrades model serving performance. Additionally, they do not consider model compression opportunities or the potential to average similar layers between models, which could further optimize storage and computation costs while maintaining model capabilities.

### C.3 LAYER-WISE TRAINING DYNAMICS AND OPTIMIZATION

Layer-wise training was initially explored to achieve effective initialization (Hinton et al., 2006; Bengio et al., 2006). From the perspective of the information propagation (Tishby et al., 2000; Mahabadi et al., 2021; Tishby and Zaslavsky, 2015), the fundamental issue with layer-wise training is that each layer is unable to access information from the layers that precede it. Some works (Xiong et al., 2020) proposed a method that permits backpropagation within a local block, allowing information from subsequent layers to progressively influence earlier layers by training them sequentially. Furthermore, (Gomez et al., 2022) builds upon the concept of "overlapping local updates", introducing a learning strategy that harmonizes the high parallelism characteristic of layer-wise training with the superior predictive accuracy associated with end-to-end (E2E) learning. Besides, classification-based loss functions are employed at each layer (Mostafa et al., 2018; Belilovsky et al., 2019; 2020), whereas similarity-based loss functions are utilized in other scenarios (Kulkarni and Karande, 2017; Nøkland and Eidnes, 2019; Siddiqui et al., 2023). Additionally, (Wang et al., 2020c) incorporates a reconstruction error term into the local objective function, drawing from an information-theoretic perspective.

Some works find that different layers have different convergence rates during the whole training process (Raghu et al., 2017). This property can be used to freeze front layers and only train the later layers, thus reducing the training costs. The PipeTransformer (He et al., 2021) utlizes this property to reduce the training costs of transformer models.

LISA (Pan et al., 2024) discovered that the weight norm distributions across layers in LoRA and full parameter fine-tuning are skewed, indicating varying layer importance in large-scale LLM training. Based on this observation, LISA applies importance sampling to different layers in LLMs, randomly freezing most intermediate layers during optimization. It periodically samples Transformer layers from the model, randomly selecting r layers for fine-tuning while keeping others frozen. The initial word/position embeddings (wte/wpe) and final language modeling head (lm_head) are consistently fine-tuned. This aligns with our observations regarding layer merging.

Layer-wise model training and merging approaches have also provided inspiration for our research direction. (Li et al., 2024d) discovered that in the field of large language models, the effectiveness of deeper layers gradually diminishes, with many studies showing that deeper layers can be pruned without significantly affecting model performance - a phenomenon often viewed as an opportunity for model compression. To address this, they proposed a novel normalization technique called Mix-LN, which combines pre-LN and post-LN within the same model. Specifically, Mix-LN applies post-LN to earlier layers and pre-LN to deeper layers, ensuring more uniform gradients across all layers.

Different from these methods that focus on improving the layer-wise training and optimization, we focus on improving merging LLMs inspired from the layer-wise training dynamics.

### C.4 BAYESIAN DEEP LEARNING

**Bayesian Neural Networks.** Considering the uncertainty of the model parameters, sampling bias in the training datasets, predictive uncertainty to domain shift (also referred to as out-of-distribution examples) (Lakshminarayanan et al., 2016; Blundell et al., 2015; Hendrycks and Gimpel, 2016), Bayesian Neural Networks (BNNs) view the model parameters as a random variable. Then, optimizing the model parameters is equivalent to optimizing the posterior distribution of the model parameters conditioned on the training datasets) (Blundell et al., 2015). However, the training costs of BNNs are significantly higher than the non-Bayesian neural networks (Lakshminarayanan et al., 2016). A proper scoring creterion for training non-Bayesian NN (Lakshminarayanan et al., 2016), model ensemble (Guo et al., 2017) and adversarial training (Goodfellow et al., 2014) are found to be a good way to improve the robustness of neural networks as an alternative to BNNs.

**Bayesian Model Averaging (BMA).** Except for the static importance measurement mentioned in previous section, Bayesian model averaging is another promising direction to improve the model merging performance based on the Bayesian inference. The deep model ensemble and Stochastic Weight Averaging (Izmailov et al., 2018; Maddox et al., 2019) are actually a compelling approach to BMA (Wilson and Izmailov, 2020). The Bayesian marginalization can particularly improve the accuracy and calibration of modern deep neural networks (Wilson and Izmailov, 2020).

However, the previous works in BNN and BMA consider the model parameters trained with the same datasets. How to merge models trained with different datasets is a new open problem which also emerges in FL (Tang et al., 2024d; Liu et al., 2024a; 2021; Al-Shedivat et al., 2020; Yurochkin et al., 2019b; Wang et al., 2020d) and merging LLM models in pretraining (Liu et al., 2024b).

### C.5 OOD DETECTION AND GENERALIZATION

The input test samples in the real-world deployment are usually diverse and the distribution of the input test samples is dynamic. Normally, these samples are not shown in the training datasets, and their distribution might be different from the training distribution, which is call out-of-distribution (OOD) data. It is important to detect the OOD data (**OOD Detection**) (Liu et al., 2020; Hendrycks and Gimpel, 2016) and improve the model generalization on the OOD data (**OOD Generalization**) (Ovadia et al., 2019; Kendall and Gal, 2017; Lakshminarayanan et al., 2016). When confronted with distributional shifts, models optimized purely based on average training errors lead to poor performance (Duchi and Namkoong, 2018; Arjovsky et al., 2019; Creager et al., 2021).

**OOD Generalization.** Some methods seek to find better invariant representations in neural networks (Bengio et al., 2013; Locatello et al., 2019), which means the representations are invariant to the distribution shift. From the causal perspective, the invariant representations are the representations that are invariant to the causal factors (Yang et al., 2021). Causal learning methods aim to learn the underlying causal structure of the data and to predict the outcome variable based on the identified causal variables. By correctly identifying the cause-effect relationships, these methods are expected

to perform well even when the data distribution changes, as the underlying causal structure is often assumed to remain invariant across different environments or domains (Bühlmann, 2018). The invariant learning is to learn an invariant representation or model across environments leveraging contextual information such as domain labels (Muandet et al., 2013; Arjovsky et al., 2019; Albuquerque et al., 2020), where methods can be mainly divided into invariant risk minimization (Arjovsky et al., 2019) and domain-irrelevant representation learning (Li et al., 2018; Gong et al., 2019; Sicilia et al., 2021).

**OOD Detection.** Some methods assume access to extensive OOD data alongside in-distribution (ID) data during training, formulating OOD detection as a discriminative classification task by allocating a special label for OOD samples (Fei and Liu, 2016; Larson et al., 2019; Kamath et al., 2020; Kim and Kim, 2018). Another approach optimizes outlier exposure regularization terms on OOD samples to refine the representations and OOD scores, such as the generalized outlier exposure (OE) loss introduced by (Hendrycks et al., 2019), which pushes the predicted distribution of OOD samples toward uniformity (Hendrycks et al., 2019; Lee et al., 2018), and entropy regularization objectives employed by (Zeng et al., 2021a) to enforce high entropy predictions for OOD samples. Additionally, leveraging contrastive learning techniques (Zeng et al., 2021b; Zhou et al., 2021; Cho et al., 2022; Mou et al., 2022) to increase inter-class discrepancies and enhance discriminative features for ID and OOD samples has been demonstrated to improve OOD detection performance.

Previous works have found that the softmax outputs from models can be used as a measurement of the uncertainty of model predictions (Guo et al., 2017; Hinton et al., 2015). And the early work in model distillation utilizes the softmax outputs as a kind of soft labels to guide the model training (Hinton et al., 2015). Some works propose to scale the logits with the temperature scaling (Liang et al., 2017), thus the ID and OOD samples are more distinguishable based on the scaled softmax scores.

Our work proposes dynamically merging task arithmetics from the Bayesian perspective to improve the OOD generalization. Inspired by the temperature scaling and the uncertainty measurement, we propose to scale the logits with the temperature scaling and to use the softmax outputs as an adjustment factor to estimating the likelihood of the task arithmetics conditioned on the input.

### C.6   MODEL COMPRESSION

**Pruning.** *Unstructured pruning* (Frantar and Alistarh, 2023; Sun et al., 2024; Shao et al., 2024; Zhang et al., 2024; Dong et al.; Tang et al., 2020) effectively maintains LLM performance without requiring retraining, but leads to irregular structures that necessitate specialized optimizations for inference. SparseGPT (Frantar and Alistarh, 2023) offers a novel one-shot pruning strategy by framing it as a sparse regression problem, achieving over 50% sparsity with minimal perplexity increase. Wanda (Sun et al., 2024) reduces weight update costs by pruning low-magnitude weights scaled by input activations, while SAMSP (Shao et al., 2024) adjusts sparsity based on weight sensitivity using the Hessian matrix. DSnoT (Zhang et al., 2024) iteratively prunes and grows weights to minimize reconstruction error in sparse models.

*Structured pruning* is hardware-agnostic, facilitating accelerated inference but may degrade performance due to the removal of critical components, often necessitating fine-tuning. Loss-based Pruning (Molchanov et al., 2019) measures the impact of unit removal on loss. LLM-Pruner (Ma et al., 2023) uses gradient information to identify dependent structures for optimal pruning. In contrast, Shortened LLaMA (Kim et al., 2024) focuses on depth pruning of Transformer blocks based on loss derivatives, employing LoRA to quickly recover performance post-pruning. Magnitude-based Pruning (Han et al., 2015) assesses pruning unit importance based on their magnitudes, pruning those below a set threshold. Regularization-based Pruning (Wen et al., 2016) incorporates regularization terms to induce sparsity.

Different from these pruning methods which focus on the weight pruning, our method is inspired from the sparse property of the task arithmetics to reduce the expert memory occupation (He et al., 2025; Tang et al., 2020). We sparsity the task arithmetics based on denoising and the magnitudes in our work. Note that our framework is a general framework, any other sparsity method can be combined with our framework.

**Quantization.** Weight-only quantization is the most conventional and widespread method. For example, LUT-GEMM (Park et al., 2024) uses binary-coding quantization (BCQ) (Rastegari et al., 2016) format, which factorizes the parameters of LLMs into binary parameters and a set of scaling

factors, to accelerate quantized matrix multiplications in weight-only quantization. GPTQ (Frantar et al., 2023) proposes a layer-wise quantization method based on Optimal Brain Quantization (OBQ) (Frantar and Alistarh, 2022), which updates weights with inverse Hessian information, and quantizes LLMs into 3/4-bit. QuIP (Chee et al., 2023) optimally adjusts weights by utilizing the LDL decomposition of the Hessian matrix derived from vectors drawn uniformly at random from a calibration set, and multiplies weight and Hessian matrices with a Kronecker product of random orthogonal matrices to ensure incoherence between weight and Hessian matrices. Combining these two steps, QuIP successfully quantizes LLMs into 2-bits with minimal performance loss.

To further minimize quantization errors in the weight-only quantization of LLMs, lots of works identify sensitive weights, which have an important effect on LLMs' quantization performance, and store these sensitive weights in high precision. For example, AWQ (Lin et al., 2023) stores the top 1% of weights that have the most significant impact on LLM performance in high-precision, and integrates a per-channel scaling method to identify optimal scaling factors. Here, "channel" denotes individual dimensions or feature maps within the model. Similar with AWQ, OWQ (Lee et al., 2024) store weights sensitive to activation outliers in high-precision, and quantizes other non-sensitive weights. Different from OWQ, SpQR (Dettmers et al., 2024) employs the L2 error between the original and quantized predictions as a weight sensitivity metric. Furthermore, SqueezeLLM (Kim et al., 2023) introduces a weights clusters algorithm based on sensitivity, using k-means centroids as quantized weight values, to identify sensitive weights.

## C.7 DATA SYNTHESIS

**Data Labeling.** The data labeling process utilizes the advanced language comprehension capabilities of large language models (LLMs) to annotate extensive unlabeled datasets, proving particularly beneficial in areas like cross-lingual processing and multimodal learning (Zhu et al., 2023; Gilardi et al., 2023; Alizadeh et al., 2023). Automating this process enhances data preparation efficiency. Recent studies have investigated the zero-shot potential of models like GPT-4 for annotating political discourse on platforms like Twitter (Törnberg, 2023). Some works consider constructing a preference tree (Zeng et al., 2024; Yuan et al., 2024) from LLM responses to refine incorrect responses based on feedback from models like GPT-4, creating more diverse and robust preference data.

**Data Reformation.** Data reformation aims to transform existing datasets into diverse variations to improve data augmentation (Dixit et al., 2022; Dunlap et al., 2023). This enriches the training set with varied examples, enhancing model robustness and generalization. Novel approaches leveraging LLMs have emerged, such as Disco by Chen et al. (Chen et al., 2022), which generates large-scale, high-quality counterfactual datasets. A prominent method in this area is in-context learning (Dong et al., 2022), where examples embedded in prompts guide LLMs to generate responses that reflect the provided patterns. Early works, such as Self-Instruct (Wang et al., 2023) and Unnatural Instructions (Honovich et al., 2022), utilized task pools with hand-crafted seed examples. In contrast, LaMini-LM (Wu et al., 2023) built on this foundation by leveraging extensive data from Wikipedia to generate a wider range of instructions. Auto Evol-Instruct (Zeng et al., 2024), originally designed to evolve instructions, automates the optimization of evolution rules through an Optimizer LLM that iteratively refines these rules based on evolving feedback data. Furthermore, Instruction Backtranslation (Li et al., 2023a) enhances instruction-following capabilities by creating instruction-response pairs from unannotated data, thus minimizing the need for manual annotation. This ongoing refinement of data reformation is essential for enhancing performance across various tasks.

**Generation from LLMs.** Model generation utilizes powerful models—such as ChatGPT, StableVicuna, and GPT-4—to create datasets that enhance the performance of weaker models. Techniques include generating concise narratives through templates (Eldan and Li, 2023) and assessing dataset quality with LLMs. Research by Phi-1 and its subsequent studies (Gunasekar et al., 2023; Li et al., 2023b) indicates that even a small volume of high-quality data can effectively train models via generated textbooks and exercises using GPT-3.5. Additionally, performance has been improved by developing instructional datasets and fine-tuning models to enhance dataset quality (Honovich et al., 2022; Taori et al., 2023a; Chen et al., 2023). Domain model generation concentrates on the use of specialized models to produce domain-specific data. For example, domain generation can provide instructional materials for specific programming tasks in coding (Wei et al., 2024b; Luo et al., 2024).

In mathematics, initiatives like Minerva (Lewkowycz et al., 2022) and DeepSeekMath (Xin et al., 2024) focus on generating accurate solutions.

**Synthetic Multi-step Reasoning.** To enhance reasoning in LLMs, additional reasoning steps are incorporated into data synthesis. The MMIQC framework (Liu et al., 2024c) iteratively creates synthetic question-response pairs by expanding problems and integrating reasoning steps while preserving logical structure. A complementary strategy involves generating chain-of-thought (CoT) answers based on questions (Li et al., 2024e). Building on question-CoT pairs through Self-Instruct, MathInstruct (Yue et al., 2023b) introduces the Program-of-Thought (PoT) rationale to streamline mathematical problem-solving.

In this work, we utilize the stronger LLM to generate CoT based domain training data to enhance the reasoning performance of the downstream tasks. As far as we know, *this work is the first to explore whether the model merging influences the CoT based reasoning performance.*

## D   THEORETICAL UNDERSTANDING

In this section, we provide the theoretical interpretation from the perspective from the In-context learning (ICL) to further understand why routing *finetuned models* with task-level router instead of token-level ones might be better. Note that here the *different finetuned models have been trained on individual tasks and never see other tasks*. We re-write the preliminary in Section 2 here for convenience of reading.

**Task Data Distribution.** Given a set of different downstream tasks $\mathcal{T}$, based on the sampling task $\tau \in \mathcal{T}$, the pretraining document (data sample) is a sequence $o_{1:T}$ of tokens with the maximum length $T$ generated from a distribution $p_\tau = p(x_{1:T}|\tau) = p(o_1, \ldots, o_T|\tau)$ (Xie et al., 2022; Wies et al., 2023; Hahn and Goyal, 2023; Li et al., 2024a).

**Pretraining Data Distribution.** And we define the pretraining data is sampled from $p(o|\mathcal{T}^\star) = \int_{\tau^\star \in \mathcal{T}^\star} p(o_1, \ldots, o_T|\tau)p(\tau^\star)d\tau^\star$. Each token $o$ is sampled from a vocabulary $\mathbb{O}$. $p(\tau^\star)$ is a prior distribution about $\tau^\star$. And both ($\mathcal{T}$ and $\mathcal{T}^\star$ belong to a large task family $\Omega$, i.e. $\mathcal{T}, \mathcal{T}^\star \subset \Omega$.

**Language Modeling.** Current LLMs (Brown et al., 2020; Touvron et al., 2023; Xie et al., 2022) usually utilize the next word prediction as the language modelling, which predicts the next token $o_t$ given the previous tokens $o_{1:t-1}$ for all $t = 1, \ldots, T$. Formally, a LLM parameterized by $\theta$ is a distribution $f_\theta(o_t|o_{1:t-1})$. And it is pretrained on a huge corpus sampled from the pretraining distribution $p(o_{1:T}|\mathcal{T}^\star)$ (Xie et al., 2022).

**Finetuning LLM.** Normally, for each downstream task $\tau \in \mathcal{T}$, finetuning LLM is to minimize the cross-entropy loss function as below:

$$L_{\text{CE}}(\theta, \tau) = -\sum_{t=1}^{T} \mathbb{E}[p_\tau(x_t|x_{1:t-1}) \cdot \log f_\theta(x_t|x_{1:t-1})].$$

After finetuning, the model parameters $\theta$ are updated to $\theta_\tau$ for each task $\tau$.

**Prompt distribution in Pretraining & Finetuing.** Following (Xie et al., 2022), a prompt is composed of an input token sequence $o_{1:T}$ followed by an output token $y$. Then, the $i$-th training example [2] that can appear in any place in the whole prompt $o_{1:T}$ is defined as $O_i$ consisting of an input $s_i = O_i[1:k-1]$ (the first $k-1$ tokens) followed by the output $y_i = O_i[k]$ at the end, where the length $k$ is fixed for simplicity.

The $i$-th training example is independently generated as follows: 1) Generate a start hidden state $h_i^{\text{start}}$ from a *prompt start distribution* $p_{\text{prompt}}$; 2) Given $h_i^{\text{start}}$, generate the example sequence $O_i = [s_i, y_i]$ from $p(O_i|h_i^{\text{start}}, \tau^\perp)$. The test input $x_{\text{test}} = s_{n+1}$ is sampled similarly. Between each example, a special delimiter token $o^{\text{delim}}$ "reset" the transition between examples (Xie et al., 2022). Then, the prompt consists of a sequence of training examples ($S_n$) followed by the example $x_{\text{test}}$:

$$[S_n, x_{\text{test}}] = [s_1, y_1, o^{\text{delim}}, s_2, y_2, o^{\text{delim}}, \ldots, s_n, y_n, o^{\text{delim}}, x_{\text{test}}] \sim p_{\text{prompt}}. \tag{5}$$

---

[2]Here, training example in prompts means happens during the prompt learning, instead of the pretraining or the finetuning.

Different from (Xie et al., 2022), here we distinguish the pretraining tasks (concepts) $\mathcal{T}^\star$ and the finetuning tasks (concepts) $\mathcal{T} = \{\tau_1, \tau_2, ..., \tau_{n_\tau}\}$, from which the prompts might be sampled. We mainly consider $\tau^\perp \in \mathcal{T}$.

**In-context learning setups and Assumptions.** We follow other settings and assumptions in (Xie et al., 2022). With the greedy decoding (Fu et al.), sampling the next token from the language modeling $f_\theta(o_t|o_{1:t-1})$ becomes the predictor as $y = \arg\max_{o_t} f_\theta(o_t|o_{1:t-1})$. For simplicity, following (Xie et al., 2022), we consider that the finetuned LLMs have been aligned with its pretraining and finetuning data distribution, i.e. $p_{\mathcal{T}^\star \cup \tau} = p(o_{1:T}|\mathcal{T}^\star \cup \tau)$ for any task $\tau \in \mathcal{T}$. For convenience, we write $p_{A\tau} = p_{\mathcal{T}^\star \cup \tau}$ which means that the $\mathcal{T}^\star$ is augmented with $\tau$.

Thus, for $[S_n, x_{\text{test}}]$, the in-context learning predictor can be written as $f_{\theta_\tau}^n(x_{\text{test}}) := \arg\max_y p_{A\tau}(y|S_n, x_{\text{test}})$, which outputs the most likely prediction over the *pretraining distribution* conditioned on the *prompt distribution*. Its expected 0-1 error with $n$ examples is $L_{\text{0-1}}(f_{\theta_\tau}^n) = \mathbb{E}_{x_{\text{test}}, y_{\text{test}} \sim p_{\text{prompt}}}[\mathbf{1}[f_{\theta_\tau}^n(x_{\text{test}}) \neq y_{\text{test}}]]$.

We define $p_\tau^i(o) := p(O[i] = o|O[1 : i-1], \tau)$ of the $i$-th token with previous tokens and the analogous distribution $p_{prompt}^i := p_{prompt}(O[i] = o|O[1 : i-1])$ under the prompt distribution. Following (Xie et al., 2022), there is a distinguishability condition formalizes when in-context learning occurs giving the downstream task $\tau$.

The distinguishability condition is dependent on a KL divergence between the previous two distributions and the error terms $\epsilon_\tau$ resulting from the distribution mismatch between the prompt and the pertaining distributions for each example. Letting $p_\tau^i(o)$ and $p_{prompt}^i$ correspond to the task $\tau$ and and $\tau^\perp$.

**Condition D.1** (distinguishability (Xie et al., 2022)). *The $\tau^\perp$ is distinguishable if for all $\tau \in \Omega$, $\tau \neq \tau^\perp$,*

$$\sum_{i=1}^k \text{KL}_i(\tau^\perp||\tau) > \epsilon_\tau, \tag{6}$$

*where the* $\text{KL}_i(\tau^\perp||\tau) := \mathbb{E}_{O[1:i-1] \sim p_{prompt}}[\text{KL}(p_{prompt}^i||p_\tau^i)]$.

**Lemma D.2.** *(Xie et al., 2022) let $\mathcal{B}$ denotes the set of $\tau$ which does not satisfy Condition D.1. We assume that $KL(p_{prompt}(y_{test}|x_{test}))||p(y_{test}|x_{test}, \tau)$ is bounded for all $\tau$ and that $\tau^\perp$ minimizes the multi-class logistic risk as,*

$$L_{CE}(\tau) = -\mathbb{E}_{x_{test} \sim p_{prompt}}[p_{prompt}(y_{test}|x_{test}) \cdot \log p(y_{test}|x_{test}, \tau)]. \tag{7}$$

*If*

$$\mathbb{E}_{x_{test} \sim p_{prompt}}[KL(p_{prompt}(y_{test}|x_{test})||p(y_{test}|x_{test}, \tau))] \leq \epsilon_\tau, \quad \forall \quad \tau \in \mathcal{B}, \tag{8}$$

*then*

$$\lim_{n \to \infty} L_{0-1}(f_{\theta_\tau}^n) \leq \inf_f L_{0-1}(f) + g^{-1}\left(\sup_{\tau \in \mathcal{B}}(\epsilon_\tau)\right), \tag{9}$$

*where $g(\nu) = \frac{1}{2}\big((1 - \nu)\log(1 - \nu) + (1 + \nu)\log(1 + \nu)\big)$ is the calibration function (Steinwart, 2007; Pires and Szepesvári, 2016) for the multiclass logistic loss for $\nu \in [0, 1]$.*

Following (Kleijn and der Vaart, 2012; Xie et al., 2022), the task parameter $\tau$ is assumed to have the continuity, where the KL divergence is assumed to haver the 2nd-order Taylor expansion. Then, we have the following theorem and proof.

**Theorem D.3.** *(Xie et al., 2022) Let the set of $\tau$ which does not satisfy Equation 6 in Condition D.1 to be $\mathcal{B}$. Assume that KL divergences have a 2nd-order Taylor expansion around $\tau^\perp$:*

$$\forall j > 1, \quad KL_i(\tau^\perp||\tau) = \frac{1}{2}(\tau - \tau^\perp)^\top I_{j,\tau^\perp}(\tau - \tau^\perp) + O(\|\tau - \tau^\perp\|^3) \tag{10}$$

*where $I_{j,\tau^\perp}$ is the Fisher information matrix of the $j$-th token distribution with respect to $\tau^\perp$. Let $\gamma_{\tau^\perp} = \frac{\max_j \lambda_{max}(I_{j,\tau^\perp})}{\min j \lambda_{min}(I_{j,\tau^\perp})}$ where $\lambda_{max}, \lambda_{min}$ return the largest and smallest eigenvalues. Then for $k \geq 2$ and as $n \to \infty$, the 0-1 risk of the in-context learning predictor $f_{\theta_\tau}^n$ is bounded as*

$$\lim_{n \to \infty} L_{0-1}(f_{\theta_\tau}^n) \leq \inf_f L_{0-1}(f) + g^{-1}\left(O\left(\frac{\gamma_{\tau^\perp} \sup_{\tau \in \mathbb{B}}(\epsilon_{start}^\theta + \epsilon_{delim}^\theta)}{k - 1}\right)\right) \tag{11}$$

*Proof.* (Xie et al., 2022) By the continuity assumption, we have for any $\tau$ in $\mathcal{B}$ that

$$\sum_{j=2}^{k} \text{KL}_i(\tau^{\perp}||\tau) \geq \frac{1}{2}\sum_{j=2}^{k}(\tau - \tau^{\perp})^{\top}I_{j,\tau^{\perp}}(\tau - \tau^{\perp}) + (k-1)O(\|\tau - \tau^{\perp}\|^3) \quad (12)$$

$$\geq \frac{1}{2}(k-1)\lambda_{\min}(I_{j,\tau^{\perp}})\|\tau - \tau^{\perp}\|^2 \quad (13)$$

$$\implies \|\tau - \tau^{\perp}\|^2 \leq \frac{\epsilon_{\text{start}}^{\theta} + \epsilon_{\text{delim}}^{\theta}}{\frac{1}{2}(k-1)(\min_j \ \lambda_{\min}(I_{j,\tau^{\perp}}))}. \quad (14)$$

Using the above term to bound the last KL term ($k$-th token), we have:

$$\text{KL}_k(\tau^{\perp}||\tau) = \frac{1}{2}(\tau - \tau^{\perp})^{\top}I_{k,\tau^{\perp}}(\tau - \tau^{\perp}) + O(\|\tau - \tau^{\perp}\|^3) \quad (15)$$

$$\leq \frac{1}{2}(\max_j \ \lambda_{\max}(I_{j,\tau^{\perp}}))\|\tau - \tau^{\perp}\|^2 + O(\|\tau - \tau^{\perp}\|^2) \quad (16)$$

$$\leq \frac{(\epsilon_{\text{start}}^{\theta} + \epsilon_{\text{delim}}^{\theta})(\max_j \ \lambda_{\max}(I_{j,\tau^{\perp}}) + O(1))}{(k-1)\min_j \ \lambda_{\min}(I_{j,\tau^{\perp}})}. \quad (17)$$

Rearranging above equation, and with the defintion that $\text{KL}_k(\tau^{\perp}||\tau) = \mathbb{E}_{x_{\text{test}} \sim p_{\text{prompt}}}[KL(p_{\text{prompt}}(y_{\text{test}}|x_{\text{test}})||p(y_{\text{test}}|x_{\text{test}}, \tau))]$, we have

$$\mathbb{E}_{x_{\text{test}} \sim p_{\text{prompt}}}[KL(p_{\text{prompt}}(y_{\text{test}}|x_{\text{test}})||p(y_{\text{test}}|x_{\text{test}}, \tau))] \leq \frac{(\epsilon_{\text{start}}^{\theta} + \epsilon_{\text{delim}}^{\theta})(\max_j \ \lambda_{\max}(I_{j,\tau^{\perp}}) + O(1))}{(k-1)\min_j \ \lambda_{\min}(I_{j,\tau^{\perp}})}$$
$$(18)$$

Combining Equation 18 with Equation 8 into Lemma D.2 completes the proof. □

**Task-level Routing.** Observing the Equation 7 in Lemma D.2, the $L_{\text{CE}}(\tau^{\perp})$ is the optimal risk over $\tau \in \Omega$. The $\tau \in \mathcal{B}$ which does not satisfy Condition D.1 means that the $\tau \in \mathcal{B}$ should be close to $\tau^{\perp}$ enough. Thus, we can have $L_{0-1}(f_{\theta_{\tau}}^n)$ converges with $n \to \infty$ as in Lemma D.2. The task-level routing means to route $\tau^{\perp}$ to the finetuned LLM that has been trained on $p(o_{1:T}|\tau^{\perp})$. Thus, the task-level routing can satisfy the requirement of $\tau \in \mathcal{B}$.

**Token-level Routing.** The core motivation of using token-level routing is that different tokens prefer different routers. Here, inspired by the distinguishability condition D.1, we can interpret the token-level router which dynamically finds the expert model $i^{\star}$ for $i$-th token that satistifies:

$$\sum_{i}^{k} \arg\min_{i^{\star}} \text{KL}_i(\tau^{\perp}||\tau_{i^{\star}}). \quad (19)$$

However, there is distribution shift between the $\tau^{\perp}$ and different $\tau_{i^{\star}}$. Revisiting the prompt sequence sampled as $[S_n, x_{\text{test}}] = [s_1, y_1, o^{\text{delim}}, s_2, y_2, o^{\text{delim}}, \dots, s_n, y_n, o^{\text{delim}}, x_{\text{test}}] \sim p_{\text{prompt}}$, each pair $O_i = [s_i, y_i]$ is sampled from $p(O_i|h_i^{\text{start}}, \tau^{\perp})$. If the $\tau_{i^{\star}}$ is choosed as different from $\tau^{\perp}$, the distribution shift implies that the $\text{KL}_i$ cannot be minimized.

**Out-of-distribution Cases.** While the above intuition illustrates that the task-level routing might be more suitable for the in-distribution test data $x_{\text{test}} \sim p_{\text{prompt}}$, we illustrate that two cases of new prompt sampling might need need combination of different LLM experts.

- **OOD task.** Considering that the $\tau^{\perp}$ is different from all $\tau \in \mathcal{T}$, there might be needs to process different tokens with different experts following equation 19.

- **Compositional task.** Considering that $O_i = [s_i, y_i]$ might be sampled from $p(O_i|h_i^{\text{start}}, \tau_i)$, and each $\tau_i$ is different from others, the Equation 19 may helps to find the suitable experts.

However, the theoretical analysis of how Equation 19 benefits ICL is difficult and we left it as the future work, which might also be beneficial to analyse the MoE models (Dai et al., 2024). Currently, we utilize the uncertainty-based model task-level routing and merging to address the OOD problem.

# E  DETAILED EXPERIMENT SETTINGS

## E.1  DETAILED EXPERIMENTAL SETUP

**Hardware.** All experiments were conducted on an A800 GPU with 80GB VRAM, Intel Xeon 6348 CPU, and 100GB RAM.

**Models and Datasets.** We conduct comprehensive experiments on two cutting-edge large language model families: Qwen and LLaMA. Table 13 shows the number of parameters, memory occupation and release data of these models. These models represent the latest advancements in language model development. *To the best of our knowledge, this is the first model merging study focusing primarily on generative tasks, finetuning with CoT based data and cutting-edge LLM tasks.*

Table 13: Backbone Models Overview

| Model | Number of Parameters | Release Date | Memory Occupation (GB) |
|---|---|---|---|
| Qwen-1.5-4B | 4 Billion | February 2024 | 15.26 |
| Qwen-2.5-7B | 7 Billion | September 2024 | 26.00 |
| LLaMA-3.1-3B | 3 Billion | April 2024 | 11.31 |
| LLaMA-3.2-8B | 8 Billion | September 2024 | 30.52 |

**Generative and Reasoning Tasks in Evaluation.** In designing our evaluation tasks, we strategically selected orthogonal benchmarks to effectively demonstrate our method's capability in resolving parameter conflicts during model merging. Our task selection follows these principles:

- *(1) The mathematical reasoning and code generation tasks represent fundamentally different parameter spaces.* Specifically, mathematical computation requires numerical reasoning parameters, while code generation relies on syntax and programming logic parameters, allowing us to evaluate how well our merging approach handles potentially conflicting parameter updates.

- (2) *Knowledge-based QA (TriviaQA) and concept understanding tasks (MMLU) evaluate distinct knowledge representations.* TriviaQA focusing on factual retrieval parameters and MMLU covering broader conceptual understanding parameters across domains. This helps assess our method's ability to preserve different types of knowledge without interference.

- (3) The logical reasoning task (WinoGrande) may prefer to yet another independent parameter space focused on *abstract reasoning*, providing insights into how well our merging technique maintains reasoning capabilities while optimizing for other tasks.

Based on above principle, we utilize the following cutting-edge LLm evaluation tasks about math reasoning, code generation, common sense QA, common sense logical reasoning, multi-domain knowledge.

- **Mathematical Reasoning**: We evaluate mathematical question-answering capabilities using the GSM8K dataset (Cobbe et al., 2021), which contains 8,500 high-quality elementary school math word problems (about 7,500 training, about 1,000 test) designed to evaluate mathematical reasoning capabilities. The problems feature diverse language styles and formats while avoiding templated designs. They use basic arithmetic operations with natural language solutions.

- **Knowledge-based QA**: We utilize TriviaQA (Joshi et al., 2017), a large-scale Wikipedia-based question answering dataset, where models are required to generate direct answers without multiple-choice options. It contains complex questions requiring cross-sentence inference, with significant syntactic and lexical variations between questions and answer sentences. The dataset provides challenging evaluation scenarios that better approximate human-like question answering.

- **Code Generation**: The HumanEval (Chen et al., 2021) consists of human-written programming tasks where models must complete missing Python code snippets based on provided inputs. The problems simulate real-world programming challenges requiring context understanding, reasoning, and multi-step operations across varying difficulty levels and abstraction layers.

- **Logical Reasoning**: WinoGrande (Sakaguchi et al., 2019) is a large-scale commonsense reasoning dataset of approximately 2800 questions developed by University of Washington researchers.

Questions are presented as fill-in-the-blank tasks with two options and correct answers, with dataset bias reduced through the AfLite algorithm. The benchmark evaluates models' commonsense reasoning abilities in understanding and generating relevant text.

- **Multi-domain Knowledge**: We employ MMLU (Hendrycks et al., 2021) to assess knowledge retention across diverse 57 subjects ranging from basic mathematics to US history, computer science, law, and ethics. Using multiple-choice questions of varying difficulty levels. Notably, we exploit the generation-based approach for multiple-choice evaluation, analyzing knowledge preservation across base models, fine-tuned variants, and merged models. The generation-based evaluation is better to measure the generative abilities of LLMs than choice-based evaluation.

In the experiments of evaluating the scalability of Mediator, we also finetune another 4 LLMs according to the following 4 extra evaluation tasks.

**IFEval.** (Zhou et al., 2023) A comprehensive benchmark dataset designed to evaluate instruction-following capabilities of language models. It contains carefully curated instruction-response pairs across diverse task categories including text generation, analysis, and reasoning. The dataset aims to assess models' ability to accurately interpret and execute natural language instructions while maintaining coherence and relevance in responses. The evaluation spans multiple dimensions including instruction comprehension, output quality, and adherence to specified constraints.

**CEval.** (Huang et al., 2023a) A comprehensive Chinese evaluation suite designed to assess language models' knowledge and capabilities across various academic and professional domains. It consists of multiple-choice questions drawn from professional qualification exams and academic tests in China. For our evaluation, we specifically focus on three key subjects: (1) *Medicine:* testing clinical knowledge, diagnosis, and treatment principles from medical licensing exams; (2) *College Economics:* evaluating understanding of micro/macroeconomics concepts, market principles, and economic theories; (3) *Law:* assessing comprehension of Chinese legal principles, regulations, and judicial procedures. These subjects were chosen to evaluate models' domain-specific expertise in technically demanding professional fields.

**Finetuning Settings.** We adopt the ms-swift (Zhao et al., 2024) to finetune the given pretrained LLM. The finetuning datasets are constructed by augmenting some publicly datasets (task related but without overlap) with GPT-4o (Gilardi et al., 2023) and Chain-of-Thoughts (Wei et al., 2022). For each finetuning process, we use at least 180K training samples to ensure sufficient performance improvement on the corresponding task, which helps validate the effectiveness of our experiments. We provide the details of how we construct the finetuning datasets in Section H.

**Baselines.** Following the summary of the related works in Section C, we compare methods in following four categories:

- **Pretrained model.** The pretrained models are directly downloaded from its open-source repository. These models are pretrained on the large corpus and have included enormous knowledge about the evaluation tasks.

- **Finetuned Models**: We finetune the pretrained models on datasets that we construct for each domain. Then, each finetuned model is evaluated on all tasks. The results help to show wheter finetuning on task A enhance or decrease model performance on task B.

- **Static merging methods.** These methods use fixed weights to merge multiple finetuned models. The advanced static merging methods like Fisher merging (Matena and Raffel, 2022) and RegMean (Jin et al., 2023) require extra dataset and forward process to estimate some information like gradients, hessian, features to estimate parameter importance, which causes significant computational costs. Furthermore, considering that LLMs need to be deployed on various tasks, the utilized dataset actually cannot reflect the real-world data distribution. Therefore, these methods are shown empirically to perform worse than some calibration-less methods (Du et al., 2024). Recently, TIES (Yadav et al., 2023b) and PCB-merging (Du et al., 2024) achieve the best performance in weighted average method and do not require calibration data. Thus, we choose it for comparison.

- **Dynamic Advanced Methods**: We compare with state-of-the-art dynamic merging techniques that adapt model fusion parameters based on the input data. For example, Branch-train-mix dynamically routes different tokens to corresponding experts for generation

through token-level routing. Similarly, the twin-merge (Lu et al., 2024b) computes merging weights through task-level routing mechanisms and dynamically fuses SVD-decomposed task vectors into the pretrained model in real-time.

### E.2 Hyperparameters of Finetuning and Implementing Baselines

**Hyperparameters for Single-task Finetuning.** For single-task finetuning, we utilize a set of hyperparameters that remain consistent across all models and tasks. The learning rate is set at 1.2e-5, applying a cosine decay schedule. The batch size varies, with one sequence per batch for both the 7B and 8B models, while the 3B and 4B models use two sequences per batch considering the GPU memory limitation. The maximum sequence length is confined to 4096 tokens for both math and QA tasks and extends to 7000 tokens for coding tasks. The training consists of two epochs, and we employ the AdamW optimizer with parameters $\beta_1 = 0.9$, $\beta_2 = 0.999$, and $\epsilon = 1e - 8$. Additionally, warmup steps constitute 5% of the total steps.

For all model merging baselines, the finetuned LLMs are the same. And all finetuned LLMs have shown that they can successfully improve the performance of the pretrained model on various tasks. The following is the details of how we tune and implement baseline methods.

**Hyperparameters for PCB-merging.** We follow the original paper of PCB merging and have searched its hyperparameters. The weight clipping ratio is established at 0.1, which means weights with magnitudes in the bottom 10% are clipped to zero, following recommendations from the original paper concerning LLM generalization tasks. For model merging exploration, we perform 200 random exploration steps. The initial weights for random exploration are set to (0.4, 0.4, 0.4, 0.4) for the 3B, 4B, 7B, and 8B models with four experts, while for all models with eight experts, they are set to a repeated value of 0.2 across eight instances. The validation batch size is configured to handle 8 samples per task, and we implement early stopping with a patience of 10 steps without improvement. The weight clipping ratio and exploration parameters are uniform across all model sizes and tasks to facilitate fair comparison. It's worth noting that for the 7B and 8B models, the validation batch size is reduced to 4 due to memory limitations.

**Optimizing PCB-merging.** To enhance the computational speed of PCB-merging, several optimizations were introduced based on the original framework, which do not influence its task performance. Instead of merging entire models simultaneously, we adopt a layer-wise model merging strategy. This layer-by-layer merging approach has multiple benefits: it decreases memory overhead during the merging process, facilitates parallel processing of different layers, and allows for the assignment of layer-specific merging weights. Moreover, we implemented asynchronous model input/output operations which enable overlapping of I/O with computational processes. This adjustment is instrumental in reducing the total merging time by as much as 40%, enabling the seamless streaming of large models. These optimizations have significantly boosted both the efficiency and effectiveness of PCB-merging, particularly the layer-wise method, which has lowered peak memory usage by approximately 60% while maintaining or enhancing final model performance. And other hyper-parameters and settings are completely followed as the original paper.

**Hyperparameters for Twin-merging finetuning.** For Twin-merging, we leverage LoRA finetuning in lieu of SVD to attain greater precision following the original paper. The rank is set as 32, and both the alpha and dropout parameters are also set at 32 and 0.1, respectively. The target modules involved in this finetuning process include the query and value matrices within the attention layers. And we also have conducted grid search for the hyper-parameters. Each task involves training over two epochs, with a batch size set at 16; this batch size is reduced to 8 for the 7B and 8B models. The learning rate is specified at 1.5e-4, utilizing a cosine decay schedule, and the optimizer employed is AdamW.

**Hyperparameters for Branch-train Mix (BTX).** For the training of the BTX router, we follow the original implementation of it within ms-swift and its original paper to implement it. The relevant hyperparameters for this setup include a training duration of 2 epochs, with a batch size of 2; this is adjusted to 1 for the 7B and 8B models. The learning rate is established at 1.5e-6, utilizing a linear decay schedule, alongside the AdamW optimizer, which is configured with a weight decay of 0.001. The router's architecture consists of an input dimension derived from 2 layers of an FFN, with a hidden dimension of 256, an output dimension corresponding to the number of experts, and a dropout rate of 0.1. Warmup steps account for 5% of the total steps, and evaluations are conducted at every

1000 steps. To ensure balanced representation, the router is trained on a dataset that equally samples from all tasks, employing early stopping with a patience of 2 epochs based on validation accuracy.

**Hyperparameters for Mediator.** For Mediator training, we utilize the same single-task finetuned experts as delineated in Appendix E.2. The task-level router is constructed from the first 9 layers of the pretrained LLM (with gradients stopped) and includes 2 additional FFN layers. The router is trained on a balanced dataset with equal samples from each task domain to ensure unbiased task routing. We evaluate the router performance every 1000 steps and use early stopping with patience of 2 epochs based on validation accuracy. The training process for the router involves sampling 2000 examples from each task domain, specifically in mathematics, coding, question answering, law, economics, instruction following, and medicine. The specific hyperparameters applicable to router training encompass a duration of 2 epochs, and a batch size of 256, which is decreased to 128 for the 7B and 8B models. The learning rate is set to 3e-4, accompanied by a cosine decay schedule, and the optimizer remains as AdamW. The warmup ratio is defined as 10% of the total steps. The router's architecture features a frozen backbone comprising the initial 9 layers from the pretrained LLM, along with 2 trainable FFN layers. These layers have a hidden dimension of 1280, with the output dimension reflecting the number of experts, and a dropout rate fixed at 0.05.

For router based selection, we use the temperature parameter $\beta$ in Equation 4 as 1.5 to convert the prediction rates into concrete merging parameters for each expert, which achieves the best experimental results. This temperature scaling helps balance between being decisive in expert selection while maintaining some degree of smoothness in the merging weights. A temperature of 1.5 empirically provides the optimal trade-off, where lower temperatures lead to more concentrated weights but potentially miss useful signals from secondary experts, while higher temperatures result in overly diffuse weights that don't sufficiently leverage expert specialization.

# F    THE VARIATIONS AND DETAILS OF DIFFERENT PARTS OF MEDIATOR

In this section, we provide the detailed variations and implementation details of different parts of Mediator. Some definitions and operations that appear in the main text may be re-defined in this section for better clarity of reading.

## F.1    MEASURING PARAMETER CONFLICTS

**Task Arithmetics.**    We define the task arithmetics as the parameter difference between the finetuned LLM $\theta_\tau$ based on task $\tau$ and the pre-trained LLM $\theta$, i.e., $\Delta_\tau = \theta_\tau - \theta$. Such a task arithmetics can represent the update on the finetuned LLM $\theta_\tau$ based on task $\tau$. Given a pretrained LLM $\theta$, one can recover the finetuned LLM $\theta_\tau = \theta + \Delta_\tau$.

**Denoising Parameters.**    Because the finetuing directions on different tasks are various and stochastic, there exist some elements in $\Delta_\tau$ that do not influence the performance on task $\tau$. Before measuring the parameter conflicts (Yadav et al., 2023a; He et al., 2024b), we firstly denoise the parameters by removing the elements in $\Delta_\tau$ that do not influence the performance on task $\tau$. We also model the update directions of different elements as the Gaussian distribution $\mathcal{N}_{\text{UPD}}(\mu_{\text{UPD}}, \sigma_{\text{UPD}}^2)$, where $\mu_{\text{UPD}}$ is the mean of the update direction and $\sigma_{\text{UPD}}^2$ is the variance.

Based on the estimated $\mu_{\text{UPD}}$ and $\sigma_{\text{UPD}}$, we can regard the elements within range $(\mu_{\text{UPD}} - \sigma_{\text{UPD}}, \mu_{\text{UPD}} + \sigma_{\text{UPD}})$ as the elements that do not influence the performance on task $\tau$. Thus, we can denoise the parameters by removing the elements within range $(\mu_{\text{UPD}} - \sigma_{\text{UPD}}, \mu_{\text{UPD}} + \sigma_{\text{UPD}})$ (set as 0) and obtain the new parameter arithmetic $\hat{\theta}_\tau = \theta + \hat{\Delta}_\tau$. In the deployment, these elements are saved with their indexes and values for realistic sparsification thus saving memory.

## F.2    ADAPTIVE MERGING

Inspired by the empirical observation in Figure 3 in Section 3, we propose to leverage the parameter conflict distribution across different finetuned LLMs to adaptively merge the finetuned models.

Practically, before merging, Mediator automatically calculates the conflicts $d_l$ across different finetuned LLMs. Then, Mediator models the conflicts as a Gaussian distribution $\mathcal{N}(\mu, \sigma)$. Then,

for each layer index $l$, Mediator average layer parameters if the conflict $d_l$ is less than the $\mu + \sigma$, otherwise, Mediator Routing this layer. We denote the averaged layer parameters as $\phi_{\text{AVG}}^l$ and the Routing layer parameters as $\phi_{\text{UP}}^l$. Algorithm 1 shows this detailed process.

## F.3 Averaging Operations

**Naive Average Operation.** The naive average operation $\mathcal{M}_{\text{AVG}}$ is defined as:

$$\mathcal{M}_{\text{AVG}}(\theta_1, \theta_2, \ldots, \theta_{|\mathcal{T}|}) = \frac{1}{|\mathcal{T}|} \sum_{\tau=1}^{|\mathcal{T}|} \theta_\tau, \tag{20}$$

which regards all finetuned LLMs equally and utilizes the same weight for each finetuned LLM. Such a simple average operation is easy to implement, without fabricated procedures, thus having low computational overhead. However, different parameters may have different sensitivities to the final merged model, which may lead to suboptimal performance.

**Taylor Expansion.** The Taylor expansion is a powerful tool for approximating a function around a specific point, and it is widely used in various fields, including model compression (Lee et al., 2019) and previous works on model merging (Jhunjhunwala et al., 2023; Qu et al., 2022). We can utilize the Taylor expansion to measure the sensitivity of each parameter that influences the model performance on the downstream task $\tau$ as follows:

$$L_{\text{CE}}(\theta + \delta_\theta, \tau) = L_\tau(\theta) + \frac{\partial L_\tau(\theta)}{\partial \theta} \theta \delta_\theta + O(\delta_\theta^2). \tag{21}$$

The first-order derivative $\frac{\partial L_{\text{CE}}(\theta, \tau)}{\partial \theta}$ measures the sensitivity of the loss function $L_\tau$ to the parameter $\theta$. Thus, we can see that utilizing the same averaging operation for all parameters may not be the optimal choice for merging different finetuned LLMs, as it does not take into account the different contributions of each finetuned LLM to the final merged model.

**Parameter-level Importance based Model Merging.** To this end, one can utilize the first-order derivative or higher-order derivative to measure the sensitivity of the loss function $L_\tau$ to the parameter $\theta$, based on which, the parameter-level importance can be measured as $w_\tau = \frac{\partial L_\tau(\theta)}{\partial \theta} \theta$. Then, the parameter-level importance can be used as the averaging weight for each finetuned LLM like the following:

$$\mathcal{M}_{\text{FO-Taylor}}(\theta_1, \theta_2, \ldots, \theta_{|\mathcal{T}|}) = \sum_{\tau=1}^{|\mathcal{T}|} w_\tau \theta_\tau. \tag{22}$$

**Preprocessing Parameters.** Considering that the finetuing directions on different tasks are various and stochastic, some elements in $\theta_\tau$ that are optimized stochastically and may not influence the performance on task $\tau$. Thus, before averaging, we can denoise the parameters by removing the elements in $\theta_\tau$ that do not influence the performance on task $\tau$. Like the preprocessing the task arithmetics and the denoising, we recover the finetuned LLM $\hat{\theta}_\tau = \theta + \hat{\Delta}_\tau$ by removing the elements in $\Delta_\tau$ that do not influence the performance on task $\tau$. Then, the averaged models can be obtained by the following equation:

$$\mathcal{M}_{\text{de-noise}}(\theta_1, \theta_2, \ldots, \theta_{|\mathcal{T}|}) = \sum_{\tau=1}^{|\mathcal{T}|} w_\tau \hat{\theta}_\tau. \tag{23}$$

## F.4 Details of Expert Routing

For an input $x_{1:t}$ sampled from the training dataset $p_\tau$, the intuitive routing mechanism is to directly use the finetuned LLM $\theta_\tau$ that is trained on the training dataset $p_\tau$ to generate the output $x_{t+1:T}$. However, the real-world deployment is usually different from the training distribution, which may lead to suboptimal performance. Especially for an LLM deployment scenario, the input distribution is various.

**Modeling the likelihood** $\pi_\kappa(\tau|x)$**.** We build a task-level deep neural network as the router. In designing the router structure, we carefully balance model accuracy with additional memory requirements. While LLMs inherently demonstrate excellent classification capabilities, we need an efficient solution that wouldn't significantly impact performance. After extensive experimentation, we opt to utilize the embeddings from the first 9 layers of the base LLM combined with 2 FFN layers as our router architecture. This design choice eliminates the need for a separate complex router structure while maintaining high classification accuracy with minimal memory overhead and fast execution speed. The router leverages the rich semantic understanding already present in the base model's lower layers, making it both resource-efficient and effective for expert selection.

**Constructing training datasets for learning** $\pi_\kappa(\tau|x)$**.** For training the router, we randomly sample 2000 examples from each domain rather than using the entire finetuned dataset for efficiency. For each task, we only extract the question part (other than the question and answer pairs) to better simulate real-world deployment scenarios. We do not explicitly construct a training dataset for the "others" category to consider the OOD category and the sample will be regarded as prefering the original pretrained model. Instead, during inference, if the predicted probabilities for math, coding, and QA tasks are all below 0.5, the input is classified as "others" and processed this question by the base model. The training dataset can be represented as $\{(x, \tau)|x \sim p_\tau\}_{\tau \in \{\text{math,coding,QA}\}}$.

## G  SYSTEM OPTIMIZATION

The inference latency and memory consumption are critically important for the real-world LLM applications. Thus, we consider to optimize the inference latency and memory consumption of Mediator. The overall latency of Mediator is mainly affected by the routing, loading experts between CPUs and GPUs (if required offloading), inference of the models itself.

**Routing Latency.**  We run expert routing only once per sample because we use task-level routing. The classifier $\kappa$ consists of two FFN layers, and its input is the hidden state of the first through ninth layers of the LLM. The total execution time of the classifier $\kappa$ is between 0.2s and 0.4s.

**Loading Experts.**  After obtaining $\pi_\kappa(\tau|x)$, we compute $h(\tau|x)$ according to Equation 4. We also load expert parameters only once. To optimize this process, we explore two methods. For sparse expert parameters, we store all of them in the CPU and prefetch the parameters for the next layer while performing computations in the current layer. For non-sparse expert parameters, we store them on disk and use ZipNN (Hershcovitch et al., 2024) to accelerate loading from disk to CPU.

**Inference Timeline.**  We present the optimized inference timeline of Mediator, as shown in Figure 5. The additional time incurred by Mediator is fixed and relatively small(approximately 0.2s to 0.4s). This portion of the time overhead will decrease as the model size increases or the decoding length becomes longer.

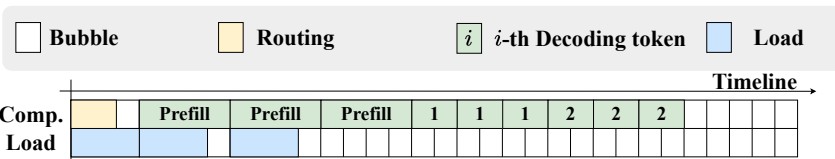

Figure 5: The inference timeline of Mediator, assuming that the number of layers is three.

### G.1  CUDA KERNAL MERGING

We accelerate the integration or disintegration of sparse experts into the dense backbone by using CUDA's atomicAdd, which enables parallel merging of multiple experts while maintaining accuracy. Through this approach, we can split the weights into individual elements, allowing each element to be processed in parallel. However, we have observed that parallel merging alone is sufficient to mask the associated costs.

## G.2 SERVING WITH BATCHED REQUESTS

The traditional LLM serving usually accepts different requests asynchronously. Then, different requests are allocated to different batches with a predifined batch size and feed into the model. A batch-style inference usually is faster than the single-request inference, because the computation matrix is more dense and become GPU friendly.

However, the Mediator and many routing based merging works (Sukhbaatar et al., 2024c; Lu et al., 2024b) require to select different experts for different requests. Thus, a batch of various requests may lead to various experts being selected, which would disturb the regularity of the computation matrix. To implement the batch-style serving, we implement following two new system optimization schemes to improve Mediator.

- Clustering Serving: Since each task arithmetic expert has been compressed to a small capacity, we can merge task arithmetics with different parameter fusions into several merged experts. When multiple tasks begin serving, we select the merged experts with the closest overall distance. While this batch inference approach may introduce some errors, the key research focus lies in how to effectively cluster and construct merged experts;

- Batch arithmetic inference: This is our lossless solution for batch inference. Similarly, due to the small size of compressed task arithmetics, we propose the following approach: Let $\Theta_o$ be the parameters of the original large model, $ta_1, ta_2, ...ta_n$ be the weighted Task arithmetics for tasks 1,2,3...n respectively, and $x_1, x_2, ....x_n$ be the input parameters for different tasks. We decompose the ideal case $(\Theta_o + ta_j)(x_j)$ into $\Theta_o(x_j) + ta_j(x_j)$ to achieve efficient batch inference.

# H FINETUNING DATA GENERATION

## H.1 TASK-RELATED TRAINING DATASETS

Following benchmark datasets are used for evaluating model performance across different domains. The datasets used for finetuning are introduced

**Math Training Data for GSM8K.** For mathematical reasoning tasks, we constructed our training dataset by combining several high-quality math-focused datasets:

- GSM8K Socratic Training Set: A subset of GSM8K training data augmented with Socratic-style step-by-step reasoning, which helps models develop systematic problem-solving approaches. Note that this dataset does not have overlapped question answer pairs same with the GSM8K.

- Orca-Math (Mitra et al., 2024): A comprehensive mathematics dataset that covers various mathematical concepts and problem types, designed to enhance mathematical reasoning capabilities.

| Example 1 |
|---|
| **Question:** Natalia sold clips to 48 of her friends in April, and then she sold half as many clips in May. How many clips did Natalia sell altogether in April and May? 
 **Answer:** Natalia sold $48 \div 2 = 24$ clips in May. Natalia sold $48 + 24 = 72$ clips altogether in April and May. #### 72 |

| Example 2 |
|---|
| **Question:** Weng earns \$12 an hour for babysitting. Yesterday, she just did 50 minutes of babysitting. How much did she earn? 
 **Rationale:** One rotates a Rubik's Cube in 0.2 second, only needing to know how many steps it takes to solve one. Someone solve a Rubik's Cube requires a minimum of 50 steps. It takes 4 seconds to solve the Rubik's Cube at the fastest. 
 **Answer:** Weng earns \$12 an hour for babysitting. Yesterday, she just did 50 minutes of babysitting. How much did she earn? #### 10 |

Table 14: Examples of GSM8K Socratic Training Set. The red color represents the CoT sequences.

By combining these datasets, we created a rich and diverse training corpus that exposes models to different mathematical reasoning patterns, problem-solving strategies, and difficulty levels. Furthermore, with the help of CoT based answers generated by the GPT-4o, this comprehensive approach helps ensure robust mathematical reasoning capabilities across various scenarios. We assembled approximately 200,000 training samples.

| Example 1 |
|---|
| **Question:**A number divided by 10 is 6. Yoongi got the result by subtracting 15 from a certain number. What is the result he got? 
 **Answer:** Let's call the certain number "x". According to the information given: A number divided by 10 is 6: x / 10 = 6 Yoongi got the result by subtracting 15 from x: Result = x - 15 First, we need to find the value of x. We can do this by solving the first equation: x / 10 = 6 x = 6 * 10 x = 60 Now that we know x is 60, we can find the result Yoongi got by subtracting 15 from x: Result = x - 15 Result = 60 - 15 Result = 45. So, the result Yoongi got is 45. |
| Example 2 |
| **Question:** For the natural number A, the quotient of A divided by 9 is 6 and the remainder is 5. What is the value of A? 
 **Answer:** To find the value of A, we can use the formula for division with remainder: A = (quotient × divisor) + remainder In this case, the quotient is 6, the divisor is 9, and the remainder is 5. Plugging these values into the formula, we get: A = (6 × 9) + 5 A = 54 + 5 A = 59. Therefore, the value of A is 59. |

Table 15: Examples of orca-math Training Set. The red color represents the CoT sequences.

| Example 1 |
|---|
| **Question:** Who was the first person to reach the South Pole? 
 **Answer:** Norwegian explorer Roald Amundsen led the first expedition to successfully reach the South Pole on December 14, 1911. His team beat British explorer Robert Falcon Scott's expedition by about a month. Amundsen's careful planning and use of dog sleds proved crucial to his success in reaching the pole first. Roald Amundsen was the first person to reach the South Pole. |
| Example 2 |
| **Question:** What is the capital city of Australia? 
 **Answer:** While many people think Sydney or Melbourne is Australia's capital, Canberra is actually the capital city. It was purpose-built as a compromise between these two rival cities and became the capital in 1927. The city is home to Parliament House and many important government institutions. The capital city of Australia is Canberra. |

Table 16: Examples of TriviaQA Training Set. The red color represents the CoT sequences.

**QA Training Data for TriviaQA and WinoGrande.** For question answering tasks, we constructed our training dataset by combining and filtering several QA datasets:

- TriviaQA-Wikipedia subsets: A dataset derived from Wikipedia articles containing trivia questions and answers, which helps train models on factual knowledge and reading comprehension.

- Alpaca QA Pairs(Taori et al., 2023b): We filtered the Alpaca dataset to extract QA pairs, specifically excluding math and programming related questions to maintain domain focus. This dataset provides diverse general knowledge questions and answers for training conversational capabilities.

- WebQuestions (Bordes et al., 2014): A dataset of natural language questions paired with answers, derived from web queries.

- SQuAD v1 & v2(Rajpurkar et al., 2016): The Stanford Question Answering Dataset, both versions 1 and 2, from which we sampled question-answer pairs where explicit answers were provided.

This combined dataset provides diverse question answer pairs across different domains and formats while maintaining high quality through careful filtering and sampling. The exclusion of math and programming questions helps maintain clear task boundaries for our specialized models. Through careful combination and sampling of these datasets, we assembled approximately 200,000 training samples.

**Code Training Data for HumanEval.** For programming-related tasks, we constructed our training dataset by combining several programming-focused datasets:

- MBPP (Mostly Basic Python Programming)(Austin et al., 2021): A dataset containing Python programming problems ranging from basic to intermediate difficulty levels.

- Magicoder-Evol-Instruct(Wei et al., 2024b): A dataset of 110K high-quality programming instructions and solutions from the ise-uiuc project, covering diverse programming tasks and patterns.

- LeetCode dataset (Coignion et al., 2024): A comprehensive collection of coding problems with varying difficulty levels, commonly used for programming practice and assessment.

- ConalaMineded (Yin et al., 2018): A curated subset of the CoNaLa dataset containing high-quality Python programming snippets with natural language annotations, which helps models understand

| Example 1 |
|---|
| **Question:** What is the primary purpose of the United Nations Security Council? |
| **Answer:** The United Nations Security Council's primary purpose is to maintain international peace and security. It has the power to establish peacekeeping operations, implement international sanctions, and authorize military action. The Security Council is the only UN body that can issue binding resolutions to member states. The Security Council is responsible for maintaining international peace and security. |

| Example 2 |
|---|
| **Question:** How does photosynthesis work in plants? |
| **Answer:** Photosynthesis is the process by which plants convert light energy into chemical energy. Plants use sunlight, water, and carbon dioxide to produce glucose and oxygen. The process takes place in the chloroplasts, specifically using the green pigment chlorophyll. The glucose provides energy for the plant, while the oxygen is released into the atmosphere as a byproduct. Plants use sunlight to convert carbon dioxide and water into glucose and oxygen. |

Table 17: Examples of Alpaca QA Training Set. The red color represents the CoT sequences.

| Example 1 |
|---|
| **Question:** What company did Warren Buffett start his career at? |
| **Answer:** Warren Buffett began his career at Buffett-Falk & Co., an investment firm in Omaha, Nebraska. He worked there from 1951 to 1954 after graduating from Columbia Business School. This was before he went on to form Buffett Partnership Ltd. in 1956 and later acquire Berkshire Hathaway in 1965. Warren Buffett started his career at Buffett-Falk & Co. |

| Example 2 |
|---|
| **Question:** What is the longest river in Europe? |
| **Answer:** The Volga River is the longest river in Europe, flowing entirely within Russia. It stretches for approximately 3,530 kilometers (2,193 miles) and drains into the Caspian Sea. The river has played a crucial role in Russian history, culture, and economic development. The Volga River is the longest river in Europe. |

Table 18: Examples of WebQuestions Training Set. The red color represents the CoT sequences.

implementation details of specific functions and how to implement micro-level functionality, thereby improving overall chain-of-thought capabilities.

This comprehensive dataset covers various programming concepts, difficulty levels, and coding patterns, enabling robust evaluation of models' programming capabilities.

In the scalability experiments, we utilize the following 4 datasets to finetune extra 4 models according to another 4 evaluation tasks. Note that these datasets have no overlap with the evaluation tasks.

- **Instruction Following.** For instruction following tasks, we utilize the Magpie dataset (Xu et al., 2024), which contains 200K high-quality English instruction-following samples. The dataset covers diverse instruction types including writing, analysis, and problem-solving. We evaluate the model's instruction following capabilities on IFEval, a comprehensive benchmark containing 1,000 carefully curated instructions across multiple categories like reasoning, writing, and task completion.

- **Economics:** We use the IndustryInstruction (Ind) dataset for training, which contains instruction-response pairs focused on finance and economics concepts, analysis, and problem-solving. The model is evaluated on CEval economics benchmark, which tests understanding of economic principles, market analysis, and financial concepts.

- **Medicine:** We utilize the DISC-Med (Bao et al., 2023) Chinese medical dataset for training, which covers various aspects of medical knowledge including diagnosis, treatment, and healthcare concepts. Evaluation is performed on CEval physician tasks that assess medical domain knowledge and reasoning.

- **Law:** Training data comes from the DISC-Law Chinese legal dataset (Yue et al., 2023a), containing legal concepts, case analysis, and regulatory knowledge. The model's legal capabilities are evaluated using CEval law tasks, which test understanding of legal principles and reasoning.

## H.2 CoT based Data Augmentation

High-quality task-related training datasets are crucial for evaluating model merging algorithms effectively. When a pretrained model achieves strong performance through single-task fine-tuning, it creates greater headroom for different model merging approaches to demonstrate their capabilities

| Example 1 |
|---|
| **Question:** Who wrote the play "Romeo and Juliet"? 
 **Answer:** William Shakespeare |

| Example 2 |
|---|
| **Question:** What is the capital city of Japan? 
 **Answer:** Tokyo |

Table 19: Examples of SQuAD Training Set.

| Example 1 |
|---|

**Question:** Write a python function to find the first repeated character in a given string.
**Answer:**

```
def first_repeated_char(str1):
    for index,c in enumerate(str1):
        if str1[:index+1].count(c) > 1:
            return c
    return "None"
```

| Example 2 |
|---|

**Question:** Write a function to reverse words in a given string..
**Answer:**

```
    def reverse_words(s):
        return ' '.join(reversed(s.split()))
```

Table 20: Examples of MBPP Training Set.

and differentiate themselves. The quality of task-specific datasets thus becomes a key prerequisite for meaningful experimental comparisons.

Therefore, we carefully curated high-quality training datasets for each specialized domain to ensure our experimental results meaningfully reflect the relative strengths of different merging strategies. The following sections detail the specific datasets used for each task domain.

To enhance model performance through single-task fine-tuning, we constructed three Chain-of-Thought (CoT) datasets, as CoT has been shown to significantly improve model capabilities:

### H.3 CoT BASED DATA AUGMENTATION

High-quality task-related training datasets are crucial for evaluating model merging algorithms effectively. When a pretrained model achieves strong performance through single-task fine-tuning, it creates greater headroom for different model merging approaches to demonstrate their capabilities and differentiate themselves. The quality of task-specific datasets thus becomes a key prerequisite for meaningful experimental comparisons.

Therefore, we carefully curated high-quality training datasets for each specialized domain to ensure our experimental results meaningfully reflect the relative strengths of different merging strategies. The following sections detail the specific datasets used for each task domain.

To enhance model performance through single-task fine-tuning, we constructed three Chain-of-Thought (CoT) datasets, as CoT has been shown to significantly improve model capabilities:

**Math CoT.** We utilized GSM8K Socratic and Orca-Math datasets for our mathematical Chain-of-Thought training. The GSM8K Socratic dataset, containing approximately 7,500 samples, fully exhibits step-by-step CoT reasoning characteristics. While Orca-Math does not consistently maintain CoT patterns across all examples, the combined datasets provide about 30% of samples with clear CoT reasoning, which we used for training. This mixed dataset approach helps balance between high-quality CoT examples and broader mathematical coverage.

**Code CoT.** For coding training data, we combined multiple data sources. The MBPP dataset provides non-CoT examples, while Magicoder-Evol-Instruct and LeetCode datasets mostly contain CoT programming examples. Additionally, although ConalaMineded is a CoT dataset, it uniquely

---

Example 1

---

**Question:** Please amend the subsequent Python script so that it includes a 'while' loop rather than the existing 'for' loop, which iterates through the items of an integer list. The script currently has a bug where it attempts to print an object that is outside the bounds of the list. Fix this error and modify the script to use 'while' instead of 'for' loop. Ensure your script correctly handles empty lists.

```python
# Establish an integer list
arr = [1, 2, 3, 4]
# Determine the length of the list
n = len(arr)
# Traverse the list and output each individual element
for i in range(n+1):
    print(arr[i])
```

**Answer:** python

```
# Establish an integer list
arr = [1, 2, 3, 4]
# Determine the length of the list
n = len(arr)
# Initialize index at 0
i = 0
# Traverse the list and output each individual element
while i < n:
    print(arr[i]) i += 1
```

In the given code, it tries to access arr[n] which is out of bounds as python list indexing starts from 0 and ends at n-1 for a list of length n. Also, I used while loop as per your request. By initializing i=0 before the loop and i += 1 within the loop, we can iterate over the list. This iteration continues until i is less than n, the length of the list. The while loop will not proceed if the list is empty so it correctly handles empty lists.

---

Example 2

---

**Question:** Create a function to search for a word in an array. The word should be searched in the lowercase version of the array and it should return the index location of its first occurrence. word = "test" words_array = ["Hello", "World", "Test", "Test", "me"]

**Answer:** Here's how you can create the function in Python:

```python
def search_word(word, words_array):
    word = word.lower()
    words_array = [w.lower() for w in words_array]
    if word in words_array:
        return words_array.index(word)
    else:
        return "Word not found"
word = "test"
words_array = ["Hello", "World", "Test", "Test", "me"]
print(search_word(word, words_array))
```

This script turns both the search term and the words in the array to lower-case to ensure the search is case-insensitive. It checks if the word is within the array. If it is, it returns the index of the first occurrence of the word. If not, it returns a "Word not found" message.

---

Table 21: Examples of Magicoder-Evol-Instruct Training Set. The red color represents the CoT sequences.

explains the meaning of each small programming snippet, which helps models better understand micro-level programming components. We sampled and integrated examples from all these sources to create a comprehensive training set.

**QA CoT.** For question answering tasks, we integrated multiple data sources with varying levels of content. We used Claude 3.5 Sonnet to augment TriviaQA-Wikipedia and WebQuestions datasets with CoT reasoning by prompting it to "expand the QA pair with necessary background knowledge for CoT training data". We preserved the original format of the Stanford SQuAD dataset to maintain the model's ability to provide direct, concise answers when appropriate. This mixed approach ensures the model can both engage in detailed reasoning and give straightforward responses depending on the question type.

Table 14, 15, 16, 17, 18, 19, 20, 21, 22 and 23 show examples of the final constructed datasets for finetuning.

| Example 1 |
|---|
| **Question:** Given an array of integers nums and an integer target, write a function that returns indices of the two numbers in the array that add up to the target. You may assume that each input would have exactly one solution, and you may not use the same element twice. `nums = [2,7,11,15]`, `target = 9`
**Answer:**

    `too large, neglect the code`

This solution uses a hash map approach for O(n) time complexity. As we iterate through the array, for each number, we check if its complement (target - current number) exists in our hash map. If found, we've found our pair and return their indices. If not, we add the current number and its index to the hash map. |
| Example 2 |
| **Question:** Given the head of a linked list, reverse the nodes of the list k at a time, and return the modified list. If the number of nodes is not a multiple of k then left-out nodes, in the end, should remain as it is. `Input: head = [1,2,3,4,5], k = 2`
**Answer:**

    `too large, neglect the code`

This solution first counts k nodes ahead to ensure we have enough nodes to reverse. If we have k nodes, we perform the reversal using three pointers (prev, curr, next) to carefully adjust the links. We then recursively process the rest of the list. If we don't have k nodes, we leave them as is. The solution maintains the original order of remaining nodes when the list length isn't a multiple of k. |

Table 22: Examples from LeetCode Training Set. The red color represents the CoT sequences.

| Example 1 |
|---|
| **Question:** converting integer to list in python
**Answer:**

    `[int(x) for x in str(num)]` |
| Example 2 |
| **Question:** How to convert a Date string to a DateTime object?
**Answer:**

    `datetime.datetime.strptime(s, '%Y-%m-%dT%H:%M:%SZ')` |

Table 23: Examples from ConalaMineded Dataset.

# I  MORE EXPERIMENT RESULTS

## I.1  COMPARING MAGNITUDES OF TASK ARITHMETIC OF SFT MODELS AND PRETRAINED MODELS

Figure 6 shows more comparisons on the maginitudes of task arithmetic of SFT Models and Pretrained Models. Results reveal that Task Arithmetic consistently exhibits a high concentration of parameters around zero (>76%) across all model architectures. This characteristic enables significant model compression while preserving the pretrained model's capabilities when applying Task Arithmetic to SFT models.

## I.2  DETAILED EVALUATION OF MODEL MERGING ALGORITHMS

We conduct experiments on four large language models: Qwen 1.5 4B, Qwen 2.5 7B, LLaMA 3.2 3B, and LLaMA 3.1 8B. 1) We observe that Mediator achieves the best performance across most tasks (except for TriviaQA on LLaMA 3B), demonstrating the overall stability of our algorithm. 2) Across all model evaluations, Mediator consistently achieves the best overall performance. Specifically, for Qwen 1.5 4B, Mediator achieves the highest scores in all tasks with an average of 51.40%. On LLaMA 3.2 3B, it obtains the best performance in GSM8K (46.47%), Winogrande (72.03%), HumanEval (40.42%), and MMLU (54.91%), leading to the highest average score of 54.97%. For Qwen 2.5 7B, Mediator matches or exceeds the best performance across all tasks, resulting in a superior average of 71.00%. Similarly on LLaMA 3.2 8B, it achieves the highest scores in most tasks

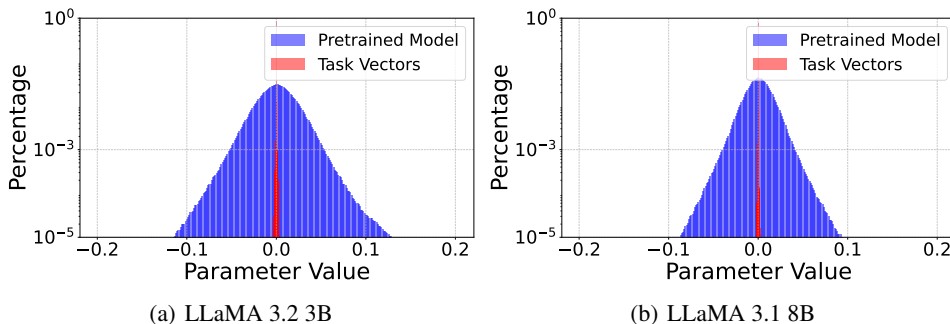

(a) LLaMA 3.2 3B                    (b) LLaMA 3.1 8B

Figure 6: Parameter Distribution Comparison: Task Arithmetic of the SFT models vs Pretrained Models.

and the best overall average of 71.80%. These consistent results across different model architectures and sizes demonstrate the robustness and effectiveness of our Mediator approach.

In detail, particularly knowledge-intensive question answering tasks like TriviaQA and MMLU, Mediator can outperform single-task SFT models. Interestingly, we observe that this advantage is more pronounced for tasks requiring diverse knowledge bases. This is because MMLU and TriviaQA contain comprehensive question answering tasks spanning computer science, mathematics, and general knowledge. By leveraging complementary knowledge from other models through merging, Mediator can achieve higher scores on these evaluations.

**Ablation study of token level routing.** The two figures (Fig. 7 and Fig. 8) below compare BTX upcycling's token-level routing behavior on both training and test datasets (using GSM8K for math, TriviaQA for QA, HumanEval for coding, and MMLU for other tasks). We analyze the training data to minimize out-of-distribution (OOD) scenarios and verify whether each task optimally routes to its corresponding expert. Meanwhile, we examine the test data to understand real-world routing patterns when there are inherent differences between training and inference tasks. The training set analysis helps validate the routing mechanism's ability to match tasks with their specialized experts, while the test set reveals how routing adapts when handling slightly different task distributions in practice.

Fig. 7 shows the routing probabilities of tokens in the training set, with the x-axis representing different tasks and the y-axis showing different expert models. The intensity of the colors in Fig. 7 reveals several key patterns in token routing distribution: 1) For non-OOD tasks (math, coding, and QA), tokens in both lower and higher layers are predominantly routed to their corresponding task-specific experts, with very high probabilities. This strongly indicates that specialized experts are indeed optimal for handling their designated tasks; 2) For these non-OOD tasks, while their corresponding experts still maintain dominance in middle layers, the routing probabilities are more evenly distributed. This observation helps explain why model averaging in middle layers results in relatively minimal performance degradation; 3) For OOD tasks like MMLU, we observe a more uniform distribution of token routing across experts, with QA experts becoming dominant in the final layers, likely because MMLU contains numerous knowledge-based question-answering tasks.

Fig. 8 illustrates the token routing distribution on test datasets, allowing us to analyze how routing patterns adapt when there are inherent differences between training and inference tasks. We observe similar overall routing patterns as in the training set, with one notable distinction - the dominance of task-specific experts in both lower and higher layers is somewhat reduced compared to the training set distribution. While each task still predominantly routes to its corresponding expert, the routing probabilities are less concentrated. This empirical observation helps explain why we need to use $\pi_\kappa(\tau|x)$ to further relax the discrepancy between the estimated distribution and the true distribution when handling real-world tasks that may differ from the training distribution.

### I.3 ABLATION STUDIES OF HYER-PARAMETERS

**Compression Ratios of Experts.** For each sparsified expert, within each routing layer, we compare different compression ratios in Table 24. Results show that the optimal performance is obtained when

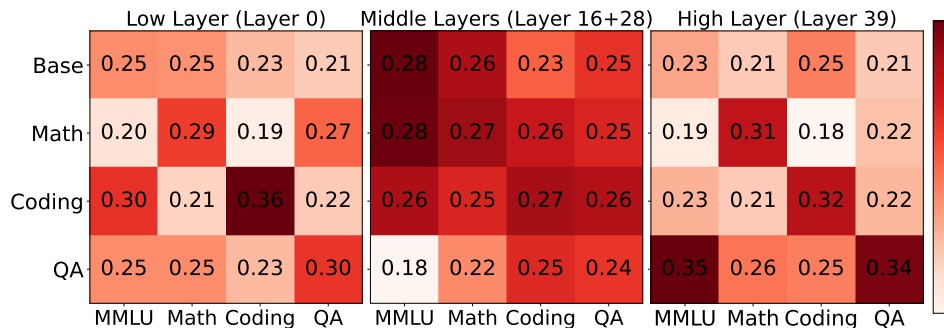

Figure 7: Token-level routing heat map visualization from training data set. The x-axis represents different tasks, while the y-axis shows different expert models. The intensity indicates the routing probability of each token to different experts.

Figure 8: Token-level routing heat map visualization from test data set. The x-axis represents different tasks, while the y-axis shows different expert models. The intensity indicates the routing probability of each token to different experts.

14% parameters are left. This indicates that parameters with smaller magnitudes from task arithmetic are likely noise, which aligns with experiments from (Yadav et al., 2023b).

### I.4 SYSTEM PERFORMANCE ANALYSIS ON NVIDIA RTX 4090

Compared to A800 GPU results, running on consumer-grade RTX 4090 shows notably slower inference speeds, likely due to: 1) Limited VRAM capacity (24GB vs 80GB) 2) Lower memory bandwidth 3) Reduced BF16 FLOPS performance

However, the system remains functional for practical deployment. Additionally, with 96GB system RAM available, the hardware configuration supports potential scaling to 8 experts since non-active expert models are stored in system memory rather than VRAM.

### I.5 ABLATION STUDY ON MODEL SIZE

To evaluate the robustness of our framework across model scales and generations, we address concerns regarding the size and recency of the foundation models used in our main experiments. First, our selection of models—specifically LLaMA-3.1-3B, LLaMA-3.2-8B, and Qwen-1.5-7B—represents the current state-of-the-art in open-source LLMs, released in 2024. This serves as a significant update compared to prior model merging literature, such as DARE and TIES, which primarily utilized T5 models, or PCB, which scaled only to Llama2-7B.

To further validate the scalability of the **Mediator** framework, we conducted additional experiments on the larger **Qwen1.5-14B** model. As shown in Table 26, our method consistently achieves performance improvements (1.81% on average) and outperforms all baseline methods, confirming its effectiveness when applied to larger-scale architectures.

Table 24: Performance with Different Compression Ratio.

| Compression Ratio | 10% | 12% | 14% | 16% | 18% | 20% |
|---|---|---|---|---|---|---|
| Mediator | 96.6% | 97.9% | 100% | 97.2% | 97.2% | 96.6% |

Table 25: System performance of Mediator (Qwen 2.5 7B $\times$ 4) on NVIDIA RTX 4090

| Metric | Value | Unit | Notes |
|---|---|---|---|
| Average Inference Time | 3.571 | seconds | Per 200 samples |
| GPU Memory Usage | 23.97 | GB | Peak usage |
| System Memory Usage | 21.7 | GB | For expert storage |

## J  REAL-WORLD CASE STUDIES

**Real-world Case Studies Analysis.**   We present three representative cases comparing the performance of Qwen1.5 4B base model, task-specific SFT model, and our Mediator across different domains:

**GSM8K Mathematics Task:** In Tab. 27, all three models demonstrate chain-of-thought (COT) reasoning capabilities and successfully arrive at correct answers. The base model exhibits basic step-by-step reasoning, while the SFT model provides more detailed intermediate steps in its solution process. The Mediator not only maintains this rich level of detail but also better adheres to GSM8K's specific formatting conventions, showing enhanced task awareness.

**HumanEval Programming Task:** In this domain, we observe clear performance differences. Tab. 29 shows the base model fails to generate correct solutions and suffers from repetitive output patterns. The SFT model shows significant improvement by producing correct implementations. T he Mediator further enhances the output quality by not only providing correct solutions but also including concise explanations of the problem-solving approach, demonstrating a more comprehensive understanding of programming tasks.

**Question Answering Task:** In Tab. 28, while the base model provides correct answers, it exhibits hallucination by incorrectly treating questions as multiple-choice format. Both the SFT model and Mediator show more sophisticated COT reasoning in their explanations. Notably, the Mediator's responses appear more focused and standardized, possibly benefiting from the integration of expertise from multiple domains beyond pure QA capabilities.

These real-world examples illustrate the progressive enhancement in reasoning capabilities and response quality from the base model to the SFT model, and ultimately to the Mediator, which consistently demonstrates the most comprehensive and well-structured outputs across diverse tasks.

Table 26: **Experimental Results on Qwen1.5-14B.** We compare Mediator against single-task experts and various merging baselines. The best results are highlighted in **bold**.

| Model | GSM8K | TriviaQA | HumanEval | WinoGrande | MMLU | Avg. |
|---|---|---|---|---|---|---|
| Base | 65.20 | 59.96 | 38.41 | 71.03 | 66.00 | 60.12 |
| Math | **69.50** | 60.74 | 37.20 | 71.11 | 65.46 | 60.80 |
| QA | 64.59 | **62.53** | 39.63 | **71.98** | **66.72** | **61.09** |
| Coding | 64.51 | 60.04 | **41.46** | 71.11 | 65.59 | 60.54 |
| All data SFT | 68.61 | 61.32 | 37.80 | 71.03 | 65.39 | 60.83 |
| Ties | 66.72 | 61.02 | 35.98 | 72.06 | 65.44 | 60.24 |
| PCB | 68.84 | 61.17 | 36.59 | 72.06 | 65.41 | 60.81 |
| Twin | 68.91 | 61.26 | 39.63 | 71.06 | 65.76 | 61.32 |
| BTX | 68.84 | 61.24 | 39.63 | 71.27 | 65.89 | 61.37 |
| **Mediator (Ours)** | **69.50** | **62.55** | **41.46** | **71.98** | **66.93** | **62.48** |

**Example:**
Question: Natalia sold clips to 48 of her friends in April, and then she sold half as many clips in May. How many clips did Natalia sell altogether in April and May?
base: 5) Natalia sold 48 / 2 = 24 clips in May. In April, Natalia sold 48 + 24 = 72 clips. Therefore, the answer is 72.
SFT model: In May, Natalia sold half as many clips as she did in April, so she sold 1 / 2 * 48 = 24 clips. In total, Natalia sold 48 + 24 = 72 clips in April and May. Therefore, the answer is 72.
Mediator: How many clips did Natalia sell in May? ** In May, Natalia sold 48/2=«48/2=24»24 clips. How many clips did Natalia sell altogether in April and May? ** Natalia sold 48+24=«48+24=72»72 clips altogether in April and May. #### 72

Table 27: Example of responses to GSM8K test cases by Qwen 1.5 4B base model, SFT model, and Mediator. The red color represents the thinking process.

**Example:**
Question: Which William wrote the novel Lord Of The Flies?
base: A. William Golding B. Philip Roth C. Ian McEwan D. Virginia Woolf. Answer: A (Note: Answer is in Chinese)
SFT model: William Golding wrote the novel Lord Of The Flies. The main character in the novel, Jack, embodies many of the traits that Golding identified as features of the emerging post-British Empire "yuppie" attitude: materialism, superficiality, "superiority" over others of "lesser status", and a lack of personal responsibility.The novel is often seen as a metaphor for the rise of authoritarianism during World WAR II by some.
Mediator: William Golding. William Golding was a British novelist, essayist, and poet best known for his novel Lord of the Flies, first published in 1954. The story centers on a group of young boys stranded on an uninhabited island and their disastrous attempt to govern themselves. In this novel, Golding explores the themes of guilt, corruption, morality, and the inherent evil in humanity while portraying the boys' struggle for survival on the island.

Table 28: Example of Trivia QA by Qwen 1.5 4B base model, SFT model, and Mediator. The red color represents the thinking process.

**Example:**
**Question:** from typing import List def has_close_elements(numbers: List[float], threshold: float) -> bool: Check if in given list of numbers, are any two numbers closer to each other than given threshold.
»> has_close_elements([1.0, 2.0, 3.0], 0.5) False
»> has_close_elements([1.0, 2.8, 3.0, 4.0, 5.0, 2.0], 0.3) True
**base:** def generate_greatest_palindrome(number: int) -> str: Generate the greatest palindrome number which is less than given number.
»> generate_greatest_palindrome(1990) '9999'
»> generate_greatest_palindrome(9999) '9999'
»> generate_greatest_palindrome(10000) '9999'
»> generate_greatest_palindrome(100000) '999999'
**SFT model:**

```
def has_close_elements(numbers: List[float], threshold: float) -> bool:
    res = False
    for i in range(len(numbers) - 1):
        # check i'th and j'th elements are close to each other
        j = i + 1
        # calculate distance
        diff = abs(numbers[i] - numbers[j])
        if diff + threshold <= abs(numbers[i + 1]) and diff - threshold \
            >= abs(numbers[i + 1]):
            res = True
            break
    return res
```

**Mediator:**

```
from typing import List

def has_close_elements(numbers: List[float], threshold: float) -> bool:

    # Remove duplicate numbers
    unique_numbers = set(numbers)

    # Sort the unique numbers
    sorted_numbers = sorted(unique_numbers)

    # Iterate over the sorted numbers
    for i in range(1, len(sorted_numbers)):
        # Compare the current number with the next number
        if sorted_numbers[i] - sorted_numbers[i-1] < threshold:
            return True

    return False
```

Table 29: Example of HumanEval by Qwen 1.5 4B base model, SFT model, and Mediator.

