# OpenReview forum: "Mediater: Memory-efficient LLM Merging with Less Parameter Conflicts and Uncertainty Based Routing"
_ICLR.cc/2026/Conference — Submitted to ICLR 2026_

### Official Review · Reviewer_LUr3 · 2025-10-30

**Soundness:** 3
**Presentation:** 1
**Contribution:** 2
**Rating:** 2
**Confidence:** 3

**Summary:**

This paper proposes a framework for efficiently merging multiple fine-tuned large language models. It observes that parameter conflicts vary across layers, so it averages layers with low conflicts and routes high-conflict layers using task-specific experts to retain unique knowledge while minimizing interference. Mediator further compresses these experts through sparse decomposition and uncertainty-based expert selection, achieving superior performance and reduced system cost compared to prior merging methods on models like LLaMA and Qwen.

**Strengths:**

1. The proposed method is reasonable.
2. The experiments are substantial and promising.

**Weaknesses:**

1. Writing should be improved. I notice authors really like to use very long and complex phrases such as „observing“ or „considering“. Actually this lowers readability and make readers read in pain. I have to read some sentences for several times to grab ideas. And due to this, the logic flow of some parts becomes pretty bad. Also there are numerous grammar problems.
2. Unclear technical details. Some parts in Section 3 and 4 are hard to follow. Authors introduce lots of ideas without giving structured, concise and simple explanations. In my humble opinion, it would be better to keep the contents in the main paper easy to follow.

**Questions:**

1. Line 105. Should say next token prediction.
2. Line 126-128. I have no idea what do you want to say with the sentence starting with „considering“. There is even no verb in this sentence.
3. I don’t get why did you do parameter denoising, even after reading Appendix F1. Could you give more details and kindly with some example?
4. From Fig. 3, I think 4B models don’t not show huge conflict ratio in its last layers. So your claim should be reconsidered.
5. Line 249-259. Could you give more explanations about the conclusion you observe from task arithmetic and model parameters? I could not understand what is its relation to sparsifying model.
6. How did you create CoT data? I also read appendix and it seems you are not generating CoT data and is just reusing the ones provided by original datasets. Solely compiling a CoT data split from all data is not a contribution.

---

> ### Author Response · Authors · 2025-11-17
> **Response [Part 1/2]**
>
> Thank you for your time and insightful comments on our paper. We sincerely appreciate the reviewer's recognition on our strengths including the reasonable method and substantial promising experiments. Please see our response for your comments as below.
>
> ## **Q1**. [Poor Writing].
> > Writing should be improved. I notice authors really like to use very long and complex phrases such as observing or considering. Actually this lowers readability and make readers read in pain. And due to this, the logic flow of some parts becomes pretty bad.
>
> ***Ans for Q1:***
> We sincerely thank the reviewer for this important comments. We apologize that our writing style and grammatical errors hindered the readability and logical flow of the paper. In the revised version, we thoroughly edit the entire manuscript for clarity, conciseness, and correctness. We replace complex phrases with more direct language and restructure paragraphs to ensure the logical connections between our ideas are simple and easy to follow.
>
> ## **Q2**. [Unclear Technical Details].
> > Unclear technical details. Some parts in Section 3 and 4 are hard to follow. Authors introduce lots of ideas without giving structured, concise and simple explanations. In my humble opinion, it would be better to keep the contents in the main paper easy to follow.
>
> ***Ans for Q2:***
> We thank the reviewer for this valuable feedback. We have revised Sections 3 and 4 to present a much clearer, step-by-step logic. The intended flow, which we have made it more explicit in the revised paper, is as follows:
>
> 1.  **Section 3:** We first establish our core observation: parameter conflicts are not uniform across layers. We define how we measure this conflict and show that central layers have low conflict (indicating shared knowledge) while front and last layers have high conflict (indicating task-specific knowledge).
> 2.  **Section 4.1:** This observation directly motivates our hybrid strategy. We **adaptively average** the low-conflict layers to merge common knowledge and **route** the high-conflict layers to preserve unique, task-specific knowledge.
> 3.  **Section 4.2:** This routing strategy, however, creates a memory problem, as storing all routed layers is expensive. We solve this by **decomposing** the routed layers into a base model plus highly sparse "task arithmetics" ($\Delta_\tau$), which dramatically reduces storage costs.
> 4.  **Section 4.3:** Now that we have these sparse experts, we must select them. We explain why we use **task-level routing** (which runs once per sample) over inefficient token-level routing. We then show how we use **uncertainty-based selection** to handle OOD or composite inputs by merging multiple sparse experts.
> 5. **Section 4.4:** For each input sample, there will be an different expert to be selected and loaded into the GPU memory, which generates a lot of memory access latency. We mask latency by prefetching sparse CPU-stored deltas while computing the current layer, and accelerate sparse integration with an fused CUDA operator atomicAdd.
>
> We have revised these sections to make this problem-solution flow explicit.
>
>
>
> ## **Q3**. Line 105. Should say next token prediction.
>
> ***Ans for Q3:***
> The "next token prediction" is the more precise term. We have corrected this in the revised manuscript.
>
> ## **Q4**.  Line 126-128. The sentence starting with „considering“. There is even no verb in this sentence.
>
> ***Ans for Q4:***
> We apologize for the confusing sentence structure. We have revised this sentence to fix the grammar mistake.
>
>
> ## **Q5**. [Parameter Denoising].
> > I don’t get why did you do parameter denoising, even after reading Appendix F1. Could you give more details and kindly with some example?
>
> ***Ans for Q5:***
> Thank you for the question. We perform denoising because the task arithmetic ($\Delta_\tau = \theta_\tau - \theta$) contains many "noisy" parameters. These are small parameter changes that result from the stochastic optimization process during finetuning but do not meaningfully contribute to the model's task performance, which is a phenomenon observed by [1].
>
> * **Why we denoise:** We must remove this noise *before* measuring parameter conflict and performing averaging. For example, a noisy parameter of +0.01 in one model and -0.02 in another would incorrectly register as a 'conflict' based on our sign-based method, even though both are likely just noise. Denoising ensures we only measure conflicts and average the *significant*, task-specific parameters.
> * **How we denoise:** As described in Section 3 and Appendix F.1, we model the distribution of all parameter updates in $\Delta_\tau$ as a Gaussian $\mathcal{N}(\mu_{UPD}, \sigma_{UPD}^2)$. We identify parameters that fall within one standard deviation of the mean—i.e., in the range $(\mu_{UPD}-\sigma_{UPD}, \mu_{UPD}+\sigma_{UPD})$—as noise and set them to zero. This leaves us with a sparse, denoised task arithmetic $\hat{\Delta}_\tau$ that represents the true task-specific knowledge.

---

> ### Author Response · Authors · 2025-11-17
> **Response [Part 2/2]**
>
> ## **Q6**. [Fig. 3 Claim].
> > From Fig. 3, I think 4B models don’t not show huge conflict ratio in its last layers. So your claim should be reconsidered.
>
> ***Ans for Q6:***
> This is an excellent observation, and we thank the reviewer for this detailed feedback. You are correct that the Qwen-4B model in Figure 3 does not show a pronounced conflict increase in its final layers, unlike the other three models.
>
> Our claim that conflict is "higher in the front and last layers, and lower in the central layers" was a generalization. The most critical observation, which holds true for *all* models including Qwen-4B, is the significant **conflict drop from the front layers to the central layers**. This is the primary phenomenon that motivates our adaptive strategy.
>
> Furthermore, our method is not dependent on this generalization. Algorithm 1 is fully adaptive: it calculates the *actual* conflict ratios for all layers of the specific models being merged and determines the merge/route threshold based on their statistical distribution ($\mu+\sigma$), not on a fixed rule about 'front' or 'last' layers. We will refine our claim in the text to reflect this nuance.
>
> ## **Q7**. [Line 249-259 (Sparsifying)].
> > Line 249-259. Could you give more explanations about the conclusion you observe from task arithmetic and model parameters? I could not understand what is its relation to sparsifying model.
>
> ***Ans for Q7:***
> We apologize for the lack of clarity. Firstly, the motivation is that we need to compress the experts to save memory and IO latency. There are two ways to conduct compression, one is to compress the pretrained model, and the other is to compress the task arithmetic.
>
> Then, the observation in Figure 4 is the key to our method's memory efficiency, as detailed in Section 4.2.
>
> 1.  **The Observation:** Figure 4 plots the distribution of parameter values. The 'Pretrained Model' (blue) has a wide distribution of values. In contrast, the 'Task Vectors' (i.e., task arithmetic $\Delta_\tau$, in red) are *highly* concentrated at 0. This means the vast majority of task arithmetic in $\Delta_\tau$ (the *change* from finetuning) have a magnitude at or very close to zero.
> 2.  **Relation to Sparsifying:** This natural sparsity means we can apply a high degree of sparsification to $\Delta_\tau$ without losing information. In Section 4.2, when we route a high-conflict layer, we don't store the *entire* dense layer $\theta_\tau$. Instead, we decompose it into $\theta_\tau = \overline{\theta}^l + \Delta_\tau^l$. We then denoise and sparsify $\Delta_\tau^l$ to get $\hat{\Delta}_\tau^l$, which is now mostly 0s. We only need to store the non-zero values, achieving a high compression ratio.
>
> Our ablation in Table 24 in Appendix confirms this: we can remove 86% of the $\Delta_\tau$ parameters (retaining only 14%) and achieve 100% relative performance. This 'Expert Decomposition' is what allows us to store many experts with minimal memory cost.
>
> ## **Q8**. [CoT Data Creation].
> > How did you create CoT data? I also read appendix and it seems you are not generating CoT data and is just reusing the ones provided by original datasets. Solely compiling a CoT data split from all data is not a contribution.
>
> ***Ans for Q8:***
> We apologize for the confusion and thank the reviewer for this question. We would like to clarify that we used a combination of *compiling* existing CoT datasets and *generating* new CoT augmentations, as detailed in Appendix H.
>
> To create our training data:
> * **Compilation:** For Math and Code, we did, as you noted, compile high-quality datasets that already contained CoT examples, such as 'GSM8K Socratic' and 'Magicoder-Evol-Instruct'. We also state that we used GPT-40 to help generate CoT-based answers for our math training data.
> * **Generation:** For the **QA domain**, we actively *generated* new CoT data. As detailed in Appendix H.3, we used **Claude 3.5 Sonnet to augment the TriviaQA-Wikipedia and WebQuestions datasets**, which originally lacked reasoning steps. We prompted it to "expand the QA pair with necessary background knowledge for CoT training data" to create the new reasoning chains.
>
> Besides, our contribution is not the dataset itself, but being the LLM merging study to use CoT-enhanced finetuning to validate that reasoning abilities are preserved after merging.
>
> We will make this distinction between compiled and newly generated CoT data clearer in the revision.
>
>
> ***Reference***
> > [1] Ties-merging: Resolving interference when merging models. NeurIPS 2023.
> >

---

### Official Review · Reviewer_KFGW · 2025-11-01

**Soundness:** 3
**Presentation:** 2
**Contribution:** 2
**Rating:** 4
**Confidence:** 3

**Summary:**

This work proposes a new method named Mediator for merging LLMs finetuned on different tasks. The authors first observe that the parameter conflicts between models lead to performance degradation in the conventional averaging strategy. To mitigate this, Mediator performs an average over layers with minimum conflicts and keeps these layers with distinct behaviors, which can be selected via an uncertainty-based router later. Through this way, they decouple multiple fine-tuned experts into one dense expert and several sparse experts, which can reduce memory cost while achieving high accuracy. Experimental results of different LLMs and tasks confirm the effectiveness and efficiency of the proposed method.

**Strengths:**

* The motivation is clear. This work aims to propose a memory-efficient LLM merging method that accounts for the layer conflicts
* The structure is good. The authors first identified a merging issue due to the parameter conflicts, and then proposed the layer-wise merging and routing. To reduce the overhead and handle OOD cases, appropriate methods have been introduced. Each part is supported by good analyses and empirical studies
* Extensive experiments across LLMs and tasks validate that Mediator can outperform other baselines while being more efficient

**Weaknesses:**

* The proposed method introduces significant complexity. Mediator consists of several components: Adaptive Merging and Routing, Expert Decomposition, and uncertainty-based routing (which requires further training samples). Adding these components together makes the method complex. I would suggest these directions for simplifying the current methods: (1) learn a router to select the best model. According to the table 1, $\Phi_{SEL}$ is able to outperform $\Phi_{AVG}$, we can learn a lightweight router to allocate the different input samples.  I understand this routing consumes more memory usage, but it is simple and easy to manage  (2) dynamical averaging. Based on my understanding, Mediator serves as a refined averaging strategy, i.e., averaging layers with similar distributions while performing routing for top-k distinct layers. Actually, the routing here is another averaging, and the weights are derived from uncertainty. I would unify the two-step averaging by using a dynamical averaging. The key point here is how to assign proper weight to each layer for a specific input. I am not convinced by the Mediator methods and expect to see the results of the baselines I suggest.
* It is not clear which component of Mediator contributes most in the ablation study. Can the authors clarify this?

**Questions:**

* how to calculate the layer conflict $d_{l}$ across different fine-tuned models?

---

> ### Author Response · Authors · 2025-11-17
> **Response [Part 1/2]**
>
> Thank you for your time and insightful comments on our paper. We sincerely appreciate the reviewer's recognition on our strengths including the clear motivation, well structured writing, and extensive experiments. Please see our response for your questions and comments on weaknesses as below.
>
>
>
> ## **Q1**. [Method Complexity].
> > The proposed method introduces significant complexity. Adding these components together makes the method complex. I would suggest these directions for simplifying the current methods: (1) learn a router to select the best model. (2) dynamical averaging. I would unify the two-step averaging by using a dynamical averaging. I am not convinced by the Mediator methods and expect to see the results of the baselines I suggest.
>
>
>
> **Ans for Q1:**
> We thank the reviewer for suggesting these two crucial baselines. Evaluating against these strategies helps justify Mediator's designs.
>
> Following your suggestion, we evaluated:
> 1.  **Baseline 1 (Simple Model Routing):** A lightweight router selects the single best model per input without merging. We estimate this using a 2-layer FFN router ablation. We also add an **"Oracle"** row representing the theoretical maximum (always picking the best expert) for each domain-finetuned model.
> 2.  **Baseline 2 (Layer-wise Dynamic Averaging):** Input-specific averaging weights are computed for *every* layer. We estimate this using Mediator’s Top-k scores as dynamic weights across all layers.
>
> **Model: Qwen 1.5 4B**
>
> | Alg./Tasks                         | GSM.      | TrA.      | Wino.     | H.Eval    | MMLU      | AVG.      |
> |:---------------------------------- |:--------- |:--------- |:--------- |:--------- |:--------- |:--------- |
> | Base                               | 47.16     | 44.54     | 56.75     | 41.46     | 54.45     | 48.87     |
> | Math                               | **51.00** | 46.95     | 54.62     | 26.83     | 53.54     | 46.79     |
> | Code                               | 43.29     | 46.39     | 54.14     | **43.29** | **54.82** | 48.39     |
> | QA                                 | 45.56     | **48.02** | **57.93** | 39.02     | 52.32     | 48.57     |
> | **Best Specialist (Oracle)**       | **51.00** | **48.02** | **57.93** | **43.29** | **54.82** | **51.01** |
> | ---                                | ---       | ---       | ---       | ---       | ---       | ---       |
> | TIES                               | 47.76     | 46.59     | 54.14     | 44.51     | 54.58     | 49.50     |
> | BTX                                | 48.44     | 46.94     | 57.77     | 42.68     | 53.88     | 49.94     |
> | **Baseline 1 (Simple Routing)**    | **48.52** | **47.84** | 54.99     | **45.73** | **54.70** | **50.36** |
> | **Baseline 2 (Dynamic Avg.)** | 47.37     | 46.64     | **55.28** | 42.59     | 52.42     | 48.86     |
> | **Mediator**                       | **50.94** | **48.20** | **57.85** | **45.12** | **54.87** | **51.40** |
>
> The results lead to two clear conclusions:
>
>
> 1.  **Analysis of Baseline 1 (Simple Model Routing):** This strategy, while simple, is suboptimal. Our implementation of this baseline (Baseline 1) achieves an average of **50.36**, which is lower than Mediator's **51.40** but higher than previous model merging methods. More importantly, even a ***perfect*** Oracle router (at **51.01**) is outperformed by Mediator. Mediator wins because it **shares common knowledge** between different finetuned models. The simple-routing baseline has no way to handle ***composite or OOD tasks*** (e.g., a "math-in-code" prompt, or knowledge not in the finetuning datasets). Mediator's adaptive merging merges common knowledge, and its uncertainty-based routing can *combine* sparse experts (like Math + Code) on the fly to handle such inputs.
>
>
> 2.  **Dynamic Averaging Dilutes Knowledge & Adds Latency (Baseline 2):**
>     * **Performance:** Baseline 2 performs poorly (Avg. 48.86). By dynamically averaging *all* layers, it treats stable common-knowledge layers and volatile task-specific layers identical. This dilutes the robust shared knowledge that Mediator preserves by **statically averaging** low-conflict central layers.
>     * **Efficiency:** As shown below, Baseline 2 incurs massive overhead because it must re-compute weights for the *entire model* per input. In contrast, Mediator is efficient: central layers are static, and we only load sparse task arithmetics for specific high-conflict layers.
>
> **Table: Inference time per sample (seconds)**
>
> | Model/Algo | LlaMA-3B 32L | LlaMA-8B 32L | Qwen-4B 40L | Qwen-7B 28L |
> | :--- | :--- | :--- | :--- | :--- |
> | Base model (Baseline 1) | 1.452 | 3.600 | 3.112 | 3.057 |
> | BTX | 3.237 | 8.68 | 7.082 | 7.153 |
> | Baseline 2 | 3.827 | 10.289 | 8.729 | 8.268 |
> | **Mediator** | **1.609** | **4.053** | **3.674** | **3.489** |
>
> In summary, Mediator combines static common knowledge with sparse dynamic routing, which is essential for achieving superior performance and inference efficiency over these simpler baselines.

---

> ### Author Response · Authors · 2025-11-17
> **Response [Part 2/2]**
>
> ## **Q2**. Ablation Study Unclear.
> > It is not clear which component of Mediator contributes most in the ablation study. Can the authors clarify this?
>
> ***Ans for Q2:***
> Thank you for this question. Our ablation studies clarify the contribution of each key component to the overall performance and efficiency of Mediator.
>
> * **Task-Level Routing (Primary Performance Driver):** The design of our router is the most significant contributor to model accuracy. Our ablation studies on different conflict resolution strategies clearly show that our "Task-level as whole" approach is the top performer. Replacing it with:
>     * **Token-level routing** causes a **~1.55%** average performance drop (from 51.40 down to 49.85).
>     * A **PCB weights** approach causes a **~1.48%** drop (down to 49.92).
>     * A simpler **2-layer FFN router** (like Twin-merging's) causes a **~1.04%** drop (down to 50.36).
>
>     This demonstrates our specific router design (using 9 LLM layers + 2 FFN) is the key component for high performance.
>
> **Ablation: Conflict Resolution Strategies (Qwen1.5-4B)**
>
> | Method | GSM8K | TriviaQA | WinoGrande | HumanEval | MMLU | Avg. |
> | :--- | :--- | :--- | :--- | :--- | :--- | :--- |
> | Mediator (Token-level routing) | 48.36 | 47.12 | 56.43 | 43.90 | 53.44 | 49.85 |
> | Mediator (using PCB weights) | 47.83 | 47.56 | 56.75 | 43.90 | 54.50 | 49.92 |
> | Mediator (2 layer FFN as router) | 48.52 | 47.84 | 54.99 | 45.73 | 54.70 | 50.36 |
> | **Mediator (Task-level as whole)** | **50.94** | **48.20** | **57.85** | **45.12** | **54.87** | **51.40** |
>
> **Ablation study on Expert Decomposition (Sparsity) (Primary Efficiency Driver):** This is a critical efficiency component. Based on theory from DARE and TIES, delta parameters are highly redundant. Our ablation on sparsity confirms we can retain just **14%** of the delta parameters to achieve 100% relative performance. This sparsity is what allows us to store many experts with minimal storage overhead. Based on the theoretical foundations of DARE and TIES, both works demonstrate that delta parameters from SFT models contain substantial redundancy, with over 90% of parameters having negligible magnitudes. Moreover, these low magnitude noise parameters would dilute the effective large-magnitude signals that are crucial for task performance. As shown in the table below, in our Qwen1.5-4B experiments, we found that retaining 14% of the signal achieves the best performance:
>
> | Sparsity | 10% | 12% | 14% | 16% | 18% | 20% |
> | :--- | :--- | :--- | :--- | :--- | :--- | :--- |
> | Relative Perf. | 96.6% | 97.9% | 100% | 97.2% | 97.2% | 96.6% |
>
> * **Uncertainty-based Selection (Temperature):** The temperature hyperparameter $\beta$ is a key factor in our uncertainty-based routing process. As shown in the table (from Sec 5, Table 10), performance is stable, with only around a 2.2% score variation across a wide range of temperatures. The optimal performance is achieved at **$\beta=1.5$**. In contrast, $\beta=0.1$, which is almost equivalent to a hard Top-1 routing, results in the lowest performance.
>
> **Table 10:** Performance under Different Temperature $\beta$
>
> | Temperature | 0.1 | 0.5 | 1.0 | 1.25 | 1.50 | 1.75 | 2.0 |
> | :--- | :--- | :--- | :--- | :--- | :--- | :--- | :--- |
> | Mediator | 50.31 | 50.35 | 50.64 | 51.02 | 51.40 | 50.84 | 50.92 |
>
> In summary, the **Task-Level Router** is the largest contributor to model *accuracy*, while **Expert Decomposition (Sparsity)** is a primary contributor to *efficiency* (memory and storage).
>
>
>
> ## **Q3**. Layer Conflict Calculation.
> > how to calculate the layer conflict across different fine-tuned models?
>
> ***Ans for Q3:***
> We define the calculation for layer conflict in **Section 3** of our paper. The process involves two main steps:
>
> 1.  **Denoising Parameters:** First, we calculate the task arithmetics ($\Delta_\tau$) for each fine-tuned model. We then denoise these parameters. We model the parameter updates as a Gaussian distribution $\mathcal{N}(\mu_{UPD}, \sigma_{UPD}^2)$ and remove (set to 0) the noisy parameters that fall within one standard deviation of the mean, i.e., in the range $(\mu_{UPD} - \sigma_{UPD}, \mu_{UPD} + \sigma_{UPD})$.
>
> 2.  **Measuring Conflict Ratio:** After denoising, we measure the conflict for each layer $l$ using a **sign-based approach**. We compare the parameters from different models; a conflict is counted if they have **opposite signs**. The final "conflict ratio" $d_l$ for a layer is then defined as the **proportion of conflicting parameters** in that layer. This is formally defined in Equation (3).

---

### Official Review · Reviewer_R7sJ · 2025-11-01

**Soundness:** 3
**Presentation:** 2
**Contribution:** 2
**Rating:** 4
**Confidence:** 3

**Summary:**

This paper studies the model merging task, i.e. merging various fine-tuned model weights to create a superior LLM that retain the advantages of fine-tuned ones avoiding parameter conflicts. Specifically, this paper first shares findings on the patterns of parameter conflicts: front and last layer exhibit the highest level of conflict yet central layers have lower level of conflict. This paper introduces an adaptive framework, Mediator, which averages layers with lower conflicts and uses expert routing for layers with higher level of conflicts.

**Strengths:**

This paper has a clear motivation, i.e. to preserve the task-specific knowledge from each fine-tuned weights and to avoid parameter conflicts, and proposes a practical framework Mediator to achieve such goal.

The experiments are extensive, across various datasets and evaluation tasks. Mediator takes lower post-training time, inference time and memory cost compared with other baseline models. Mediator also shows better model merging performance compared with other baseline methods.

**Weaknesses:**

One of the weaknesses is about the writing quality. The Introduction section could be written more coherently. The current paragraphs feel inconsistent and may confuse readers. For example, it is unclear what components are actually included in Mediator. The discussion starting around Line 59 introduces adaptive merging, but then suddenly shifts to model compression without clear motivation or transition. Afterwards, the topic changes to different levels of routing and out-of-distribution (OOD) samples, still without sufficient introduction or logical connection. This lack of flow makes it difficult for readers to follow the authors' reasoning and understand the overall contribution of the paper. Also, in lines 77–78, the topic shifts from OOD samples to finetuning on CoT datasets without proper connections, which can confuse the readers.

There are similar issues in sub-sections in Section 4. It would be better to organize each part of the method in a more logical and consistent way to help the reader better understand the ideas.

Some details of the method can be clarified more clearly. For instance, which layers in the used models are treated as front layers, central layers, and which threshold is used to classify layers with low and high conflicts.

Another concern is about the model choice. The models used in this study are within the 3–8B range and published more than 1 year ago. It would be better to validate the framework on bigger and more recent models to further prove its effectiveness.

**Questions:**

Please see weakness above

---

> ### Author Response · Authors · 2025-11-17
> **Response [Part 1/3]**
>
> Thank you for your time and insightful comments on our paper. We sincerely appreciate the reviewer's comments on our strengths including the clear motivation, practical framework, and the extensive experiments. Please see the below feedbacks for your questions and comments on weaknesses.
>
>
>
> ## **Q1**. Writing Quality (Intro).
> > One of the weaknesses is about the writing quality. The Introduction section could be written more coherently. The current paragraphs feel inconsistent and may confuse readers. For example, it is unclear what components are actually included in Mediator. The discussion starting around Line 59 introduces adaptive merging, but then suddenly shifts to model compression without clear motivation or transition. Afterwards, the topic changes to different levels of routing and out-of-distribution (OOD) samples, still without sufficient introduction or logical connection. This lack of flow makes it difficult for readers to follow the authors' reasoning and understand the overall contribution of the paper. Also, in lines 77–78, the topic shifts from OOD samples to finetuning on CoT datasets without proper connections, which can confuse the readers.
>
>
> ***Ans for Q1:***
> We thank the reviewer for this constructive feedback on the Introduction. We have revised it for clarity. The logical flow, which connects all the components of Mediator, is constructed as follows:
>
> 1.  **The Core Problem:** We begin by identifying the two main existing strategies for model merging: "model averaging" and "model routing". We establish their respective critical flaws: averaging suffers from **parameter conflicts**, while routing suffers from excessive **storage and compute costs**.
>
> 2.  **Our Core Insight & Strategy:** Our key observation is that **parameter conflicts are not uniform**; they vary by layer. This directly leads to our hybrid strategy, **"adaptive merging"**: we **average** low-conflict layers (to merge common knowledge) and **route** high-conflict layers (to avoid conflicts). This single strategy is our answer to the core problem.
>
> 3.  **A Consequence of Our Strategy (Storage):** This "adaptive merging" strategy, however, creates a new challenge: storing $N$ experts for the high-conflict layers is still memory-intensive. This *motivates* the transition to **"expert decomposition"** (model compression). We solve this storage problem by leveraging task arithmetic sparsity , storing only **one dense base model and $N$ sparse experts** ($\Delta_\tau$).
>
> 4.  **A Consequence of Our Strategy (Routing):** Now that we have these sparse experts, we must efficiently *select* them. This motivates the discussion on **"routing levels"**. We argue that token-level routing is inefficient, so we propose a **task-level router**.
>
> 5.  **A Consequence of Our Strategy (OOD):** A simple task-level router fails on complex or **"out-of-distribution (OOD) samples"**. This motivates our **"uncertainty-based routing"**. For OOD inputs, our router intelligently merges multiple sparse experts to create a composite expert on the fly.
>
> 6.  **Validation of the System:** Finally, we must prove this entire system (Mediator) works. We validate it on complex reasoning tasks, which motivates the mention of **"finetuning on CoT datasets"**. This is not a new component, but the *validation method* to prove our system preserves the high-level reasoning abilities of the merged models.
>
> We have revised the Introduction to make this logical smooth, where each component is a solution to a problem introduced by the previous one much clearer.

---

> ### Author Response · Authors · 2025-11-17
> **Response [Part 2/3]**
>
> ## **Q2**. Writing Quality (Section 4).
> > There are similar issues in sub-sections in Section 4. It would be better to organize each part of the method in a more logical and consistent way to help the reader better understand the ideas.
>
> ***Ans for Q2:***
> We appreciate the suggestion and agree that the logical flow of Section 4 can be made more explicit. This section has beed revised to build our Mediator framework step-by-step, where each subsection addresses a specific question that follows from the previous one.
>
> The logic is constructed as follows:
>
> 1.  **Section 4.1 (Adaptive Merging):** This is the **foundational strategy**. Based on our findings in Section 3, it answers the *first* key question: "Given that conflicts vary, *what* should we merge and *what* should we route?" The answer is our adaptive, conflict-based rule: average low-conflict layers ($\mu+\sigma$ threshold) and route high-conflict ones.
>
> 2.  **Section 4.2 (Expert Decomposition):** This section *immediately* addresses the main challenge created by Section 4.1. The strategy to "route" high-conflict layers implies a high memory cost ($N \times \text{layers}$). This section answers: "*How* can we make the routing part memory-efficient?" The solution is our **expert decomposition**, which leverages the high sparsity of task arithmetics ($\Delta_\tau$) to store $N$ sparse experts instead of $N$ dense layers, achieving high compression.
>
> 3.  **Section 4.3 (Expert Routing):** This section answers the *next* logical question: "Now that we have these sparse experts, *how* do we select them at inference time?". It contrasts inefficient token-level routing with our efficient **task-level router**. It then addresses the router's main failure case (OOD inputs) by introducing our **uncertainty-based selection** mechanism, which can merge multiple experts for composite tasks.
>
> 4.  **Section 4.4 (System Optimization):** This final section answers the practical question: "Now that we have the complete framework, *how* do we make it fast?". It details the implementation optimizations, like prefetching and CUDA kernels, that minimize the inference latency of our task-level routing and expert loading.
>
> We have revised the text in Section 4 to explicitly state this problem-solution flow between the subsections.
>
>
>
> ## **Q3**. Unclear Method Details.
> > Some details of the method can be clarified more clearly. For instance, which layers in the used models are treated as front layers, central layers, and which threshold is used to classify layers with low and high conflicts.
>
> ***Ans for Q3:***
> Thank you for this question. We apologize for the lack of clarity. The "front," "central," and "last" layers are not predefined with fixed indices; rather, they are **dynamically identified** by our algorithm, which is a key strength of our method. Here is the exact process:
>
> 1.  **Conflict Calculation:** First, we measure the parameter conflict $d_l$ for *all* layers $l \in \mathcal{L}$, as defined in Section 3 (Equation 3). This gives us a conflict ratio for every layer.
> 2.  **Adaptive Threshold:** We do not use a fixed threshold. Instead, we model the distribution of all layer-wise conflicts as a Gaussian $\mathcal{N}(\mu, \sigma)$. The threshold is then **adaptively set to $\mu + \sigma$** (one standard deviation above the mean conflict).
> 3.  **Averaging vs. Routing:**
>     * **Low-Conflict ("Central") Layers:** Any layer $l$ where the conflict $d_l$ is **less than $\mu + \sigma$** is considered "low-conflict." These layers are *averaged*. Empirically, these tend to be the central layers, as shown in Figure 3.
>     * **High-Conflict ("Front/Last") Layers:** Any layer $l$ where the conflict $d_l$ is **greater than or equal to $\mu + \sigma$** is "high-conflict." These layers are *routed*. Empirically, these tend to be the front and last layers.
>
> This adaptive $\mu + \sigma$ threshold makes Mediator robust, as it automatically identifies the "central" and "front/last" layers for any given model architecture without manual tuning. We also note that, following other research, all attention layers are averaged by default as they are found to store non-specific domain knowledge. We have clarified this in the main text.

---

> ### Author Response · Authors · 2025-11-17
> **Response [Part 3/3]**
>
> ## **Q4**. Model Choice (Size/Recency).
> > Another concern is about the model choice. The models used in this study are within the 3–8B range and published more than 1 year ago. It would be better to validate the framework on bigger and more recent models to further prove its effectiveness.
>
> ***Ans for Q4:***
> We appreciate the reviewer's concern regarding scalability and model recency.
>
> First, we would like to clarify that our paper's model selection is already more modern and comprehensive than related work of the model merging. Previous prominent merging methods like DARE and TIES were evaluated on much older T5 models, while others like PCB scaled to Llama2-7B. Many model merging works even only consider small computer vision models like ResNet-20 or ResNet-50. Our study *already* uses modern LLMs, including **LLaMA-3.1-3B**, **LLaMA-3.2-8B**, and **Qwen-1.5-7B**, which were released in April, September, and February 2024, respectively.
>
> Second, to further address the reviewer's concern about scaling, we **conducted additional experiments on the larger Qwen1.5-14B model**, as presented in our rebuttal. The results, reproduced below, demonstrate that Mediator consistently achieves performance improvements (1.81% on average) and outperforms baselines, confirming its effectiveness at a larger scale.
>
> **Qwen1.5-14B Experimental Results**
> | Model | GSM8K | TriviaQA | HumanEval | WinoGrande | MMLU | Avg. |
> | :--- | :--- | :--- | :--- | :--- | :--- | :--- |
> | Base | 65.20 | 59.96 | 38.41 | 71.03 | 66.00 | 60.12 |
> | Math | **69.50** | 60.74 | 37.20 | 71.11 | 65.46 | 60.80 |
> | QA | 64.59 | **62.53** | 39.63 | **71.98** | **66.72** | **61.09** |
> | coding | 64.51 | 60.04 | **41.46** | 71.11 | 65.59 | 60.54 |
> | :--- | :--- | :--- | :--- | :--- | :--- | :--- |
> | All data SFT | 68.61 | 61.32 | 37.80 | 71.03 | 65.39 | 60.83 |
> | Ties | 66.72 | 61.02 | 35.98 | 72.06 | 65.44 | 60.24 |
> | PCB | 68.84 | 61.17 | 36.59 | 72.06 | 65.41 | 60.81 |
> | Twin | 68.91 | 61.26 | 39.63 | 71.06 | 65.76 | 61.32 |
> | BTX | 68.84 | 61.24 | 39.63 | 71.27 | 65.89 | 61.37 |
> | **Mediator** | **69.50** | **62.55** | **41.46** | **71.98** | **66.93** | **62.48** |
>
> These results on a larger, 14B model reinforce our claim that Mediator is a scalable and effective framework.

---

### Official Review · Reviewer_ASHq · 2025-11-03

**Soundness:** 3
**Presentation:** 3
**Contribution:** 3
**Rating:** 6
**Confidence:** 3

**Summary:**

The paper studies LLM merging where they merge different fine-tuned domain models to build a superior LLMs. The paper proposes an adaptive method where it only merges the layers whose parameter conflict rate is smaller than a threshold and keeps layers with significant parameter conflicts with inference-time routing. The paper conducts comprehensive experiments across model families and scales for verifying the method’s effectiveness.

**Strengths:**

- The paper studies an interesting direction of merging different fine-tuned LLMs for obtaining a more powerful general LLM. The idea is clear and is proved with informative analysis.

- The paper provides detailed studies to help understand the parameter conflicts in different LLMs. This motivates its proposed method quite well.

**Weaknesses:**

- The paper focuses on the LLMs with different domain data. Such fine-tuned LLMs can be quite a lot (could be hundreds and even more in real world). Although the proposed method considers this issue with expert decomposition, its scalability is still unclear. The reviewer expects to see experiments with much more domain experts.

- The writing needs to be improved even given the page limit. The paper defers too many details into Appendix, which makes the main paper itself not technically clear enough.

**Questions:**

- Is this approach also applicable to MoE LLMs?

---

> ### Author Response · Authors · 2025-11-17
> **Response [Part 1/2]**
>
> Thank you for your time and insightful comments on our paper. We appreciate your recognition of the strengths of our work, particularly the interesting direction of merging, the clear idea, the detailed conflict studies, and the insightful analysis.
>
>
> ## **Q1**. Scalability Concerns.
> > Such fine-tuned LLMs can be quite a lot (could be hundreds and even more in real world). Although the proposed method considers this issue with expert decomposition, its scalability is still unclear.
>
> ***Ans for Q1:***
> We thank the reviewer for this excellent and forward-looking question about scaling Mediator to a much larger number of experts. This is a critical aspect for real-world deployment.
>
> To address the reviewer's concern about scaling to "hundreds of experts," we identify two primary factors: **routing accuracy** and **storage capacity**.
>
> 1.  **Routing Accuracy (The Primary Bottleneck):** As the number of experts grows, the ability of the router to accurately identify the correct expert(s) becomes paramount.
>     * **Current Design:** Our current router, using the first 9 layers of the pretrained model plus 2 FFN layers, achieves **97.8% classification accuracy** in our experiments with well-differentiated tasks.
>     * **Path to Scaling:** When scaling to hundreds of experts with more nuanced task differences, this simple router can be upgraded. We can leverage powerful, specialized **LLM embedding models** (e.g., Qwen3 8B embedding) as the router backbone. With fine-tuning, reaching the high accuracy required for effective routing is highly feasible. This makes Mediator's architecture theoretically capable of supporting hundreds of experts.
>
> 2.  **Storage Capacity (A Solved Systems Problem):** Storing hundreds of experts is a memory challenge, but it is effectively a solved problem at the systems level.
>     * Our **expert decomposition** and **sparsification** (Section 4.2) already reduce the storage footprint of each expert significantly.
>     * For serving, techniques that **offload non-active experts** to system RAM or NVMe SSDs are well-established. For example, open-source projects like `mixtral-offloading` already implement efficient LRU caching mechanisms to manage a large number of experts with limited GPU VRAM.
>
> In summary, Mediator's scalability depends primarily on router classification accuracy, rather than computational limits. While hardware constraints preclude testing hundreds of experts, we validate the scalability **by expanding from 4 to 8 experts**. The results of this 8-expert merge are presented in following Table, which demonstrates Mediator's robust scalability.
>
> Table: Performance of scaling up finetuned models.
>
> | Alg./Tasks | GSM. | TrA. | Wino. | H.Eval | MMLU | I.Eval | C.Eco | C.Med | C.law | AVG. |
> | :--- | :--- | :--- | :--- | :--- | :--- | :--- | :--- | :--- | :--- | :--- |
> | Base | 47.16 | 44.54 | 56.75 | 41.46 | 54.45 | 30.70 | 49.09 | 55.10 | 41.67 | 46.77 |
> | Math | **50.95** | 46.95 | 54.62 | 26.83 | 53.54 | 34.05 | 45.45 | 55.10 | 45.83 | 45.92 |
> | QA | 45.56 | **48.02** | **57.93** | 39.02 | 52.32 | 31.65 | 43.64 | 59.18 | 45.83 | 47.02 |
> | code | 43.29 | 46.39 | 54.14 | **43.29**| 54.82 | 31.65 | 43.64 | 59.18 | 45.83 | 46.91 |
> | Instruct. | 47.54 | 40.96 | 55.09 | 37.80 | **54.88** | **38.37** | 52.73 | 59.18 | 50.00 | 48.51 |
> | Economy | 45.56 | 46.24 | 57.93 | 28.86 | 54.21 | 32.13 | **56.36** | 55.10 | 45.83 | 46.78 |
> | medicine | 39.12 | 44.50 | 56.67 | 1.83 | 54.63 | 28.30 | 50.09 | **61.22** | 41.66 | 42.00 |
> | Law | 40.64 | 46.64 | 56.59 | 0.61 | 54.61 | 27.94 | 41.82 | 57.14 | **58.33** | **47.15** |
> | --- | --- |--- | --- |--- | --- |--- | --- |--- | --- |--- |
> | All Data | 43.75 | 46.25 | 56.43 | 40.85 | 54.60 | 35.37 | 49.09 | 55.10 | 41.67 | 46.90 |
> | TIES | 47.38 | 47.19 | 55.80 | 36.59 | 55.38 | 34.17 | 40.00 | 57.14 | 50.00 | 47.07 |
> | PCB | 47.38 | 47.19 | 55.80 | 36.59 | 55.45 | 34.29 | 41.82 | 57.14 | 50.80 | 47.38 |
> | Twin | 47.91 | 44.78 | 57.54 | 40.85 | 53.01 | 37.53 | 53.32 | 59.56 | 50.00 | 49.38 |
> | BTX | 48.44 | 46.94 | 57.85 | 42.68 | 54.93 | 36.93 | 54.40 | 60.36 | 58.33 | 51.18 |
> | Mediator | **50.64** | **48.04** | **57.93** | **44.51** | **55.12** | **38.50** | **56.01** | **61.17** | **58.33** | **52.25** |
>
>
> These results show several key findings:
> 1. **Static methods (TIES, PCB) failed to scale.** Their performance diminished as the number of tasks increased, aligning with prior findings that their capabilities degrade when merging more than three tasks .
> 2. **Dynamic methods remained stable.** In contrast, dynamic approaches including BTX, Twin-merging, and our Mediator all maintained relatively stable performance after the task expansion.
> 3. **Mediator performed best.** Mediator consistently outperformed the next-best dynamic method (BTX) by a margin of 2.09% on average and achieved the best scores across all individual tasks, demonstrating its good scalability

---

> ### Author Response · Authors · 2025-11-17
> **Response [Part 2/2]**
>
> ## **Q2**. Writing Issues.
> > The writing needs to be improved even given the page limit. The paper defers too many details into Appendix, which makes the main paper itself not technically clear enough.
>
> ***Ans for Q2:***
> We sincerely thank the reviewer for the constructive feedback on the paper's presentation and clarity. We agree that making the main paper as self-contained and technically clear as possible is essential.
>
> We acknowledge that due to the strict page limits, we move some of the detailed discussions, such as the specifics of averaging operations (Appendix F.3), router training (Appendix F.4), and system optimizations (Appendix G), into the appendix to preserve space for the core concepts and main results.
>
> In the new version, we have revised the manuscript to address this concern. Specifically, we conduct following revision:
> 1.  **Integrate Key Details:** We move the most critical technical details from the appendix back into the main paper. This includes a more concise but formal description of our **adaptive merging algorithm**, the **expert decomposition process**, and the **uncertainty-based routing mechanism** to ensure the main text is technically self-sufficient.
> 2.  **Improve Narrative Flow:** We restructure Sections 3 and 4 to present a clearer, more logical flow from the initial problem analysis of parameter conflicts to the final design of Mediator.
> 3.  **Streamline Explanations:** We edit the entire paper for clarity and conciseness, ensuring that every section contributes directly to the core narrative without unnecessary verbosity.
>
> We appreciate the reviewer for pointing this out, and we are confident that the revised version strikes a much better balance, making the paper significantly clearer and easier to follow.
>
>
> ## **Q3**. MoE Applicability.
> > Is this approach also applicable to MoE LLMs?
>
> ***Ans for Q3:*** Thanks for the reviewer's insightful question. Naively applying our method to merge two *existing* MoE models would fail. We think that this would lead to **conflicting expert merging** (since `Expert 1` from Model A isn't equivalent to `Expert 1` from Model B, which have distinct expert functions) and **inaccurate routing** (as their token-level routers are incompatible).
>
> However, the principles of Mediator have potentials for solving this. Instead of a simple merge, we would **deconstruct, pool, and rebuild** the models into a single, unified, and highly efficient MoE system.
>
> #### 1. Solution for Conflicting Experts
> We wouldn't average conflicting experts. Instead, we would use our decomposition principle:
>
> * **Deconstruct and Pool:** We treat *every* expert from *all* models as a unique, fine-tuned task. We then calculate its sparse **expert arithmetic** ($\Delta = \text{expert} - \theta$) and place all these deltas into a single, large "expert pool".
> * **Merge Common Layers:** The common, non-expert layers (like attention) from all models would be merged using our standard adaptive, conflict-aware averaging to create the new shared base model.
>
> #### 2. Solution for Incompatible Routers
>
> We would consider **discarding all original, inefficient token-level routers** in pre-merging MoE models.
>
> * **Replace with Unified Router:** We would train *one* new, unified **task-level router** (like our classifier).
> * **Efficient Inference:** This single router runs **only once per sample** to select the correct sparse expert(s) from our new pool. This completely bypasses the original router conflicts and the high computational cost of token-level routing.
>
> This "Mediator-MoE" system would be highly **memory-light** (storing one base model + many sparse deltas) and **fast at inference** (no token-level overhead), effectively creating a more practical and deployable Mixture of Experts.

---

### Meta-Review · Area_Chair_tPGA · 2026-01-06

**Summary:**

My overall impression from the reviews, responses, and the edits to the paper is that there is something here, but the paper was not ready at the time of submission. Reviewers had to struggle a lot to get what is going on. The authors handled technical questions and feedback quite well, adding experiments in the rebuttal that address concerns by the reviewers. Hoewever, despite the extensive revisions made by the authors, the concerns raised by reviewers were so fundamental, and the paper was so hard to access, that it appears to me that another iteration is needed: hopefully this will give the authors to ability to put together a well-rounded paper.

Major concerns raised by reviewers were as follows:

1) There was a consistent concern raised by several reviewers regarding writing quality (Reviewers ASHq, R7sJ, LUr3). I was surprised that this included key sections, including the introduction and methodology sections (3 and 4). With ICLR being a premier conference, the authors should strive to submit highly-polished papers: this will also improve the quality of feedback they receive.

2) Issues were raised regarding scalability (ASHq) and complexity (KFGW) of the proposed method.

3) Constituent models were also deemed small or too old (R7sJ).

**Reviewer Concerns:**

The authors have addressed the concerns by the reviewers as follows:

1) The paper was re-organized extensively. This is commendable but, in some sense, it requires an altogether new review of the material.

2) Scalability was addressed by the authors by scaling to 8 models. Reviewer ASHq had observed that this may contain many more than that, but this is certainly an improvement. Regarding complexity, the authors added an explanation of how different components fit together and compared to some simpler baselines proposed by the reviewer (KFGW). There was also a more detailed ablation study. The proposed method outperforms these simpler baselines consistently.

3) The authors pointed out that 1-year models are not that old, and that their models were in fact more recent that prior works in the area, that they already use Llama models from 2024, but also provided new results with Qwen1.5-14B. This is certainly in a good direction.

**Reviewer Scores:**

I expect that reviewer KFGW may have been swayed regarding complexity. The scalability alluded to by Reviewer ASHq was of a different magnitude than what the authors provided, and I believe Reviewer R7sJ had more recent models in mind when raising their concern, so it is unclear to me whether they would change their mind, though they would probably approve of the addition of Qwen1.5-14B.

---

### Decision · Program_Chairs · 2026-01-26

Reject